# Predicting the effect of habitat modification on networks of interacting species

Phillip P.A. Staniczenko[1,2,3], Owen T. Lewis[4], Jason M. Tylianakis [5,6], Matthias Albrecht[7], Valérie Coudrain[8], Alexandra-Maria Klein[9] & Felix Reed-Tsochas[3,10]

A pressing challenge for ecologists is predicting how human-driven environmental changes will affect the complex pattern of interactions among species in a community. Weighted networks are an important tool for studying changes in interspecific interactions because they record interaction frequencies in addition to presence or absence at a field site. Here we show that changes in weighted network structure following habitat modification are, in principle, predictable. Our approach combines field data with mathematical models: the models separate changes in relative species abundance from changes in interaction preferences (which describe how interaction frequencies deviate from random encounters). The models with the best predictive ability compared to data requirement are those that capture systematic changes in interaction preferences between different habitat types. Our results suggest a viable approach for predicting the consequences of rapid environmental change for the structure of complex ecological networks, even in the absence of detailed, system-specific empirical data.

[1] National Socio-Environmental Synthesis Center (SESYNC), Annapolis, MD 21401, USA. [2] Department of Biology, University of Maryland College Park, Maryland, MD 20742, USA. [3] CABDyN Complexity Centre, Saïd Business School, University of Oxford, Oxford OX1 1HP, UK. [4] Department of Zoology, University of Oxford, South Parks Road, Oxford OX1 3PS, UK. [5] Centre for Integrative Ecology, School of Biological Sciences, University of Canterbury, Christchurch 8140, New Zealand. [6] Department of Life Sciences, Imperial College London, Silwood Park Campus, Ascot SL5 7PY, UK. [7] Institute for Sustainability Sciences, Agroscope, Zurich 8046, Switzerland. [8] Mediterranean Institute of Marine and Terrestrial Biodiversity and Ecology, Aix-Marseille University, University of Avignon, CNRS, IRD, IMBE, Marseille 13284, France. [9] Chair of Nature Conservation and Landscape Ecology, Faculty of Environment and Natural Resources, University of Freiburg, Freiburg D-79106, Germany. [10] Oxford Martin School, University of Oxford, Oxford OX1 3BD, UK. Correspondence and requests for materials should be addressed to P.P.A.S. (email: pstaniczenko@sesync.org)

Anthropogenic land-use intensification reduces habitat complexity, with profound consequences for plant and animal species[1]. The most immediate effects of habitat simplification are shifts in the frequency and specificity of interactions between consumer and resource species[2]. These shifts result in changes to weighted network structure[3, 4] and can have significant practical consequences, as species interactions underpin crucial ecosystem services such as biological control, pollination and seed dispersal[5–8]. Field studies have begun to quantify how interaction frequencies differ among habitat types[9–13], but exhaustive collection of these data can be laborious and a bottleneck to understanding community responses to environmental changes, especially for species-rich communities containing rare and undocumented species[11]. Models that could predict interaction frequencies in modified habitats would help alleviate this problem, but several hurdles need to be overcome[14–16]. In particular, some changes to interaction patterns will result simply from changes in random encounter rates when species' abundances change, whereas others will result from altered foraging behaviour. It would be useful to describe how changes in relative species abundance[17] vs. changes in species behaviour[18] contribute to changing network structure. Furthermore, it is important to describe these changes at the level of individual field sites, and not just for aggregated networks built from interaction data collected across multiple field sites. Separating relative species abundance and species behaviour is important because large differences in recorded interaction frequencies can be attributed to random encounter among species even when there are large differences in relative species abundance[19], without the need to appeal to more complex ecological processes or mechanisms[20, 21]. In other cases, assuming only random encounters may be insufficient to fully explain changes in weighted network structure, so by separating out the contribution of relative species abundance it will be easier to identify and investigate the effects of habitat modification on species behaviour. Such clarifications are especially relevant for understanding major structural alterations of a habitat, such as deforestation: in addition to changes in relative species abundance, predator foraging efficiency and strategy are affected by decreases in habitat complexity[22] and prey switching, in turn, depends on resource availability and accessibility[23].

In this study, we test whether we can accurately predict the effects of habitat modification on the structure of weighted host-parasitoid networks[10–13] (parasitoids are insects that live in or on the body of their host, eventually killing it). Our approach involves networks sampled at field sites in both modified and relatively unmodified habitat types (hereafter 'unmodified habitat types'), and uses mathematical models that both estimate differences in relative species abundance between field sites and separate random-encounter effects from differences in species behaviour. We represent the combined effect of host and parasitoid species behaviour by interaction preferences. Interaction preferences were originally designed to improve measurements of nestedness in weighted networks[24]; here, we use them to describe differences in species behaviour between field sites in similar and different habitat types, and to make predictions of weighted network structure. We hypothesise that species behaviour does not change significantly between field sites in similar habitat types but does change significantly between field sites in different habitat types. This hypothesis would correspond to small differences in individual interaction preferences between field sites in similar habitat types, with larger differences between field sites in different habitat types. It also suggests that predicting weighted network structure at new field sites in a similar habitat type to existing data should be more straightforward than if new field sites are in a different habitat type.

Because interaction preferences may change as habitats are modified, we focus on predicting weighted network structure in modified habitat types using models primarily calibrated with data collected from unmodified habitat types. We consider a total of seven models with different complexities and data requirements, and show that neglecting to separate changes in relative species abundance from changes in species behaviour results in poor predictions of weighted network structure. We then assess the performance of five models based on ecological mechanisms that do make this separation and show that including increasingly more information from modified habitat types results in progressively better predictions. We find that models that capture systematic, community-wide changes in interaction preferences offer the best combination of model complexity and performance. These changes could, for example, relate to altered resource selectivity by consumers in habitat types with minimal forest coverage. Our new modelling approach represents a simple yet powerful way of scaling up existing data to predict weighted network structure across multiple field sites of a given habitat type, predict the effects of habitat modification, and inform the amount and type of additional data that should be collected in novel environments to improve predictions.

## Results

**Weighted host-parasitoid networks.** Weighted networks record the frequency of interactions between pairs of species in a community and have become the standard tool for studying changes in interspecific interactions. We tested the performance of our models using empirical networks from four independent studies: Ecuador[10], Indonesia[11], Swiss lowland[12] and Swiss meadow[13]. These studies involve similar guilds of interacting species (cavity-nesting bees and wasps and their parasitoid consumers), but are drawn from diverse ecosystems, including tropical forest and agroforest, temperate meadows and plains, as well as modified versions of these habitat types. In each study, interaction data were collected from modified and unmodified habitat types, with coordinated sampling at multiple field sites in a given habitat type (see Methods section and Supplementary Note 1).

We analysed each study as a separate data set and organised data into a three-level hierarchy: network, group of networks and data set. Each network was built from interaction data collected at a single field site. For mathematical convenience, we represent weighted networks as matrices with entries $B_{ijk}$ that record the number of interactions, also referred to as counts, between host species $i$ and parasitoid species $j$ at field site $k$. To test hypotheses more easily, we grouped networks by habitat type and used metadata to identify two features with each group: habitat complexity and consumer-resource ratio. For habitat complexity, we labelled groups as either 'forested' or 'open' based on metadata including tree species richness and measurements of light intensity at ground level. We defined consumer-resource ratio as the total number of successful parasitism events across all species divided by the total number of parasitised and unparasitised hosts collected in the field. This measure indicates how easily parasitoids are able to locate their hosts in particular habitat types, and we labelled groups as being associated with either 'high' or 'low' consumer-resource ratio.

Across the four data sets, we considered 12 groups and situated them in quadrants defined by habitat complexity and consumer-resource ratio (Fig. 1 and Supplementary Table 1). These quadrants represent different categories of relative habitat modification, which allowed us to test whether host and parasitoid species behaviour—represented by interaction preferences—changes between field sites in similar or different habitat

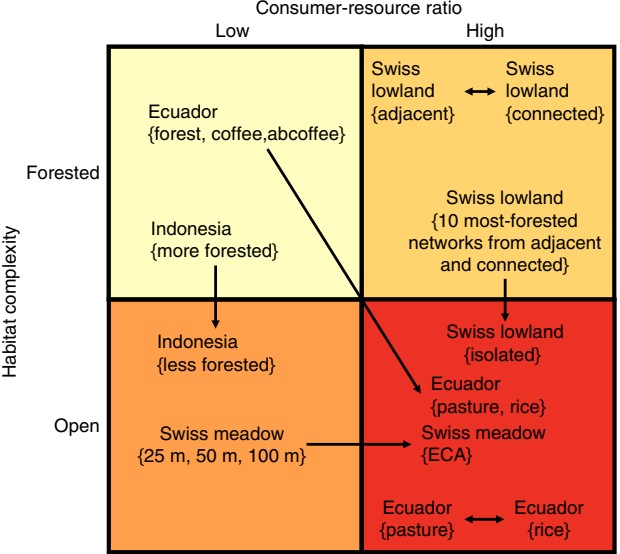

**Fig. 1** Groups of networks organised by habitat type and arranged according to relative levels of habitat modification. We considered a total of 12 groups across four host-parasitoid data sets and used metadata to identify two features with each group: habitat complexity (forested or open) and consumer-resource ratio (low or high, indicating how easily parasitoids are able to locate their hosts). Arrows within a quadrant represent predicting weighted network structure between similar habitat types in the same data set; and arrows between quadrants represent predicting weighted network structure between different habitat types in the same data set, with the direction pointing from unmodified-to-modified habitat types

types, and what effect such changes may have on the predictability of weighted networks. We tested for changes in species behaviour between field sites in similar habitat types by assessing model predictions between groups from the same data set in the same quadrant (e.g., pasture and rice in Ecuador). To test for the effects of habitat modification, we used predictions between groups from the same data set but in different quadrants (e.g., forest and rice in Ecuador). Using consumer-resource ratio as an additional dimension of habitat modification allowed us to analyse the effects of restoring intensively managed meadows as ecological compensation areas in the Swiss meadow data set, which contains networks only from open habitat types.

**Interaction preferences**. Interaction preferences describe how counts in a weighted network deviate from their expected values according to the assumption of random species encounter[24]. Under this assumption, the expected number of counts between two interacting species is proportional to the product of their relative abundances, and so random species encounter is synonymous with a mass action process[17]. Network data, however, often do not include independent measurements of species abundance or local population density. But given sufficient interaction data, it is possible to estimate relative species abundances that are consistent with mass action (see Methods section). In this way, weighted network structure can be decomposed as $B_{ijk} \propto \gamma_{ij}\hat{x}_i\hat{x}_j$; where $\gamma_{ij}$ is a contribution due to interaction preferences, and $\hat{x}_i\hat{x}_j$ is a contribution due to random species encounter where $\hat{x}_i$ and $\hat{x}_j$ are estimated or effective abundances of host and parasitoid species, respectively. Entries in a preference matrix $\gamma$ have value $\gamma_{ij} = 1$ if an interaction is consistent with mass action; $\gamma_{ij} > 1$ if counts are higher than expected, corresponding to a preferred interaction; and $\gamma_{ij} < 1$ if counts are lower than expected, corresponding to a less-preferred

interaction[24]. Forbidden interactions[20] have $\gamma_{ij} = 0$, and arise, for example, if a host species has evolved an immune response to prevent successful parasitism by a particular parasitoid species[25]. This decomposition of $B_{ijk}$ assumes that a single preference matrix is valid across all field sites included in the set of $k$-indices under consideration, e.g., across all networks in the same group. This assumption is useful for prediction because a single, model-generated preference matrix can then be used to determine weighted network structure at multiple field sites.

We refer to $\hat{x}_i$ and $\hat{x}_j$ as effective abundances because they can be considered a functional property of the system that contributes directly to recorded interaction counts, and also because their values may be different from other estimates of species abundance, such as those obtained from survey data. As explained in Methods section, the above decomposition assumes that effective abundances hold across all field sites in the same habitat type. This is a necessary assumption because there are often insufficient data in individual networks to determine non-trivial estimates of relative species abundance at individual field sites, and explains why there is no $k$-index attached to effective abundances despite it being mathematically more desirable.

It is worth emphasising the importance of modelling species abundances at the level of individual field sites, even if it is necessary to assume the same value for effective abundance at multiple field sites. This is because representing system properties using spatially aggregated data can give misleading results. For example, consider five networks that each contain the same two host ($i = 1$ and $i = 2$) and singe parasitoid species. When aggregated, there are 15 counts to the first host and 10 counts to the second host; and we therefore estimate relative host species abundance as $X_{i=1} = 15$ and $X_{i=2} = 10$. However, following our suggested approach, we find that effective abundances are $\hat{x}_{i=1} = 3$ and $\hat{x}_{i=2} = 6$ for the two host species. In this example, the relative magnitudes of host species abundance at the level of individual field sites are the reverse of the estimate based on aggregated data. This is because, on closer inspection, we might find that the number of recorded counts at the five field sites is something like {3, 3, 3, 3, 3} for the first host and {10, 0, 0, 0, 0} for the second host, which means that the second host ($i = 2$) should really be modelled as being more abundant than the first host ($i = 1$). In general, using the sum of counts across networks as a proxy for abundance or population density will underestimate values for spatially less-common species. Similar issues arise with temporal data aggregation[26]. We discuss the related topic of aggregating networks by species taxonomy in Supplementary Note 1.

Before showing how preference matrices can be used as predictive models, we first summarise how interaction preferences derived from empirical data differ between groups of networks in the same data set (see also Supplementary Notes 2 and 3 and Supplementary Table 2). When comparing like-for-like entries in the two preference matrices associated with a pair of groups, we found that a large fraction of interaction preferences changed significantly even between similar habitat types (Ecuador: 30%; and Swiss lowland: 28%; there were insufficient data to perform the analysis with the other two data sets). Less surprisingly, a greater fraction changed significantly between unmodified and modified habitat types (Ecuador: 47%; Indonesia: 20%; Swiss lowland: 36%; and Swiss meadow: 33%). A greater proportion of interaction preferences changed significantly for incumbent interactions (those observed in both groups) than for switches (interactions observed in only one of the two groups). Among incumbent interactions, there were more significant increases in interaction preference than decreases; there was no pattern with switches.

**Table 1 Seven models for predicting weighted network structure at new field sites in a novel environment**

| Model | Description | Data requirement | Application |
|---|---|---|---|
| Null | Biologically plausible interactions at a field site occur with the same frequency | Presence or absence of an interaction at a field site in the novel environment | Reference predictions that assume recorded interaction counts are uninformative |
| Aggregate counts | Recorded interaction frequencies are informative at all other field sites without additional data processing | Weighted interaction networks from multiple field sites not in the novel environment | Reference predictions that assume recorded interaction counts have intrinsic predictive value |
| Random encounter | Interaction frequencies are proportional to the product of host and parasitoid species' abundances | Relative species abundance in the novel environment | Reference predictions for a minimally complex mechanistic model |
| Alternative preferences | Species-level processes and other ecological mechanisms do not change between different environments | Relative species abundance in the novel environment and existing network data to derive a preference matrix | Predicting between similar habitat types |
| Correlated preferences | Altered resource selectivity by parasitoid species (consumers) based on habitat complexity | Relative species abundance, an existing preference matrix and a known general pattern for reordering entries according to the level of habitat complexity in the novel environment | Predicting between different habitat types |
| Specified preferences | New parasitoid species (consumer) foraging strategies in the novel environment | Relative species abundance, an existing preference matrix and a subset of network data from the novel environment on the interactions involved in new foraging strategies | Predicting between different habitat types |
| Complete characterisation | Species behaviour is so complex that all interaction preferences must be individually characterised in the novel environment | Relative species abundance and weighted interaction networks from multiple field sites in the novel environment | Reference predictions for a maximally complex mechanistic model |

Models are ordered from top-to-bottom by increasing model complexity and amount of data required for calibration

**Predicting network structure using interaction preferences**. In addition to analysing interaction preferences derived from network data, we can also use them to make predictions of weighted network structure. For a group of networks, we predicted weighted network structure at a new field site as

$$B^*_{ijk} \propto \gamma_{ij}\hat{x}_i\hat{x}_j \qquad (1)$$

where $\gamma_{ij}$ is an element from a preference matrix generated by a predictive model, and $\hat{x}_i$ and $\hat{x}_j$ are effective abundances at the new field site (of course, if species abundances are known at the field site then those values could be used instead of $\hat{x}_i$ and $\hat{x}_j$).

Testing this approach using our data sets involved five steps. First, we selected a calibration and test group from the same data set. Second, we inferred effective abundances from interaction data in the test group to represent values at the new field site. Third, we used a predictive model to generate a preference matrix based primarily on information from the calibration group. Fourth, we combined the effective abundances with the preference matrix to produce a predicted set of interaction counts (Eq. 1). Fifth, we assessed model performance by comparing the predicted distribution of counts among species to the recorded distribution in the test group. These steps were repeated for each pair of calibration and test groups.

The simplest model in this approach, the random encounter model, assumes very limited species behaviour such that all interactions at a field site are indistinguishable from mass action. All entries in this model's corresponding preference matrix have value $\gamma_{ij} = 1$ if an interaction is not forbidden, and zero otherwise. By contrast, the most complex model, the complete characterisation model, assumes that changes in species behaviour are so elaborate that each interaction preference in the matrix must be characterised individually using data from the habitat type of a new field site.

In between the two modelling extremes, we designed the alternative preferences model for predicting between similar habitat types. This model assumes that species behaviour changes very little between similar habitat types, and so the preference matrix derived from one group of networks (the calibration group) is useful for predicting weighted network structure at a new field site. We developed two further models for predicting between different habitat types. The correlated preferences model assumes that parasitoid selectivity for hosts is more pronounced in open compared to forested habitat types. This model is based on our observation that, in open habitat types, if a host species was involved in a high-preference interaction then its other interactions usually had much lower preference, leading to significant negative correlations between the preferences of individual interactions and the average preferences of neighbouring interactions (see Supplementary Note 4 and Supplementary Fig. 1). Even after accounting for such systematic differences between preference matrices following habitat modification, prediction may be limited due to new consumer foraging strategies[18, 22, 23] or as yet unidentified processes between interacting species. The specified preferences model accounts for this possibility by 'hardcoding' entries for influential interactions in preference matrices. This model is based on our observation that only a small fraction of interactions need to be characterised in modified habitat types to predict almost all changes to weighted network structure. These influential interactions did not only correspond to numerous recorded counts (Supplementary Fig. 2), as might be expected, but did typically involve abundant and generalist host and parasitoid species (Supplementary Fig. 3). For reference, the full set of models and their data requirements are summarised in Table 1.

We modelled switches (interactions present in the test group but not calibration group) in two ways: (i) switches follow mass action; or (ii) switches are inherently less-preferred interactions (Methods section). Assuming mass action switches consistently led to better model performance, so we present those results only (it is worth noting, however, that some switches had interaction preferences that differed significantly from the mass action value of one, see Supplementary Table 2 and Supplementary Note 3).

**Assessing model performance**. We quantified the accuracy of model predictions using a likelihood function based on the multinomial distribution[20] (Eq. 2 in Methods section). We chose this likelihood function because it describes how well a model is able to explain the recorded distribution of interaction counts among species at a field site. However, comparing likelihoods across field sites and data sets is not straightforward because likelihood will scale with the sum of counts in a network, which naturally varies among field sites. As such, we compared model performance among field sites using the measure $\mathcal{F}_{M,k}$ (Eq. 3 in Methods section), which rescales the likelihood of model $M$ at field site $k$ by the likelihood of a null model that assumes all non-forbidden interactions are equally likely to be observed. In general, models performed less well at field sites with very few recorded counts (Supplementary Fig. 4). This was due to the limited possibility for non-random and ecologically meaningful weighted structure to be observed in networks built using small amounts of interaction data.

For a given model, we found that $\mathcal{F}_{M,k}$ varied greatly among networks in the same group, which was potentially masking meaningful differences in model performance (Supplementary Fig. 5). This variation was due, in part, to our use of a single preference matrix to predict weighted network structure at all field sites in a group (Eq. 1). So to better compare model performance, we also used the measure $\mathcal{R}_M$ (Eq. 4 in Methods section), which describes model performance at the group level. This measure still compares predicted to recorded counts at individual field sites, but involves calculating likelihood for all field sites in a group at once. With $\mathcal{R}_M$, the likelihood of model $M$ is rescaled to the likelihood of the simple random encounter model (corresponding to $\mathcal{R}_M = 0$) and the likelihood of the maximally complex complete characterisation model (corresponding to $\mathcal{R}_M = 1$).

**Predicting between similar habitat types**. The alternative preferences model performed well when calibration and test groups were in similar habitat types (for both modified–modified and unmodified–unmodified combinations of groups). With the Ecuador data set, $\mathcal{R}_M = 0.8$ when using interaction data from pasture sites to predict weighted network structure at rice sites ($\mathcal{R}_M = 0.82$ when predicting pasture using rice); and with the Swiss lowland data set, $\mathcal{R}_M = 0.59$ and $\mathcal{R}_M = 0.68$ when predicting between two groups of forested habitat type (log-likelihoods in Supplementary Table 3). Therefore, simply combining an existing preference matrix with abundance data from a given location can be useful for predicting network structure when species behaviour is not expected to change at a new field site.

**Predicting between different habitat types**. Conventional analyses implicitly assume that recorded counts have intrinsic

**a** Aggregate counts

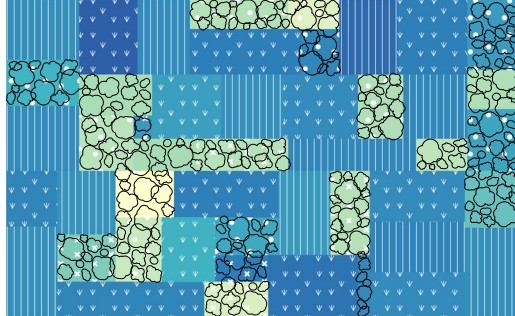

**b** Random encounter

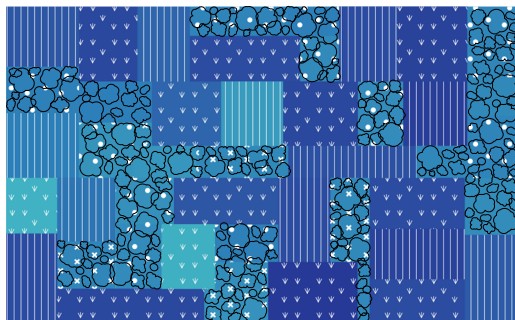

**c** Correlated preferences

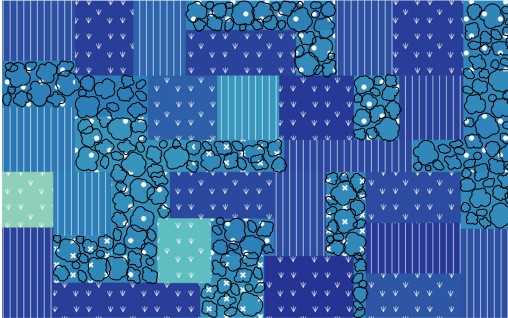

**d** Specified preferences

Rice   Pasture   Coffee agroforest   Abandoned coffee agroforest   Forest

−15      0      0.8   0.9  1.0
Model performance ($\mathcal{F}_{M,k}$)

**Fig. 2** Model performance at predicting weighted network structure at field sites in modified and unmodified habitat types in Ecuador. Field sites are represented by rectangles, with patterns indicating habitat type; rectangle areas are proportional to the number of successful parasitism events recorded at each field site. The different models (**a–d**) represent different ecological mechanisms and were calibrated using interaction data collected from field sites in unmodified, forest and agroforest habitat types and used to predict weighted network structure at field sites in modified, pasture and rice habitat types; and vice versa (a full list of models is in Table 1). Colours indicate the likelihood of models rescaled to a null model ($\mathcal{F}_{M,k} = 0$) that assumes all interactions have the same probability of being recorded (see Eq. 3 in Methods section). Notice the large increase in model performance when moving from the aggregate counts model (**a**), which does not separate relative species abundance from interaction preferences, to the random encounter model (**b**), which does. The smaller differences in model performance among the random encounter, correlated preferences (**c**) and specified preferences (**d**) models are assessed further in Fig. 4. The alternative preferences and complete characterisation models are omitted because their performances for this data set are similar to the random encounter and specified preferences models, respectively

predictive value, such that interaction data from one habitat type can be used to make predictions at field sites in other habitat types without additional data processing. In this vein, the

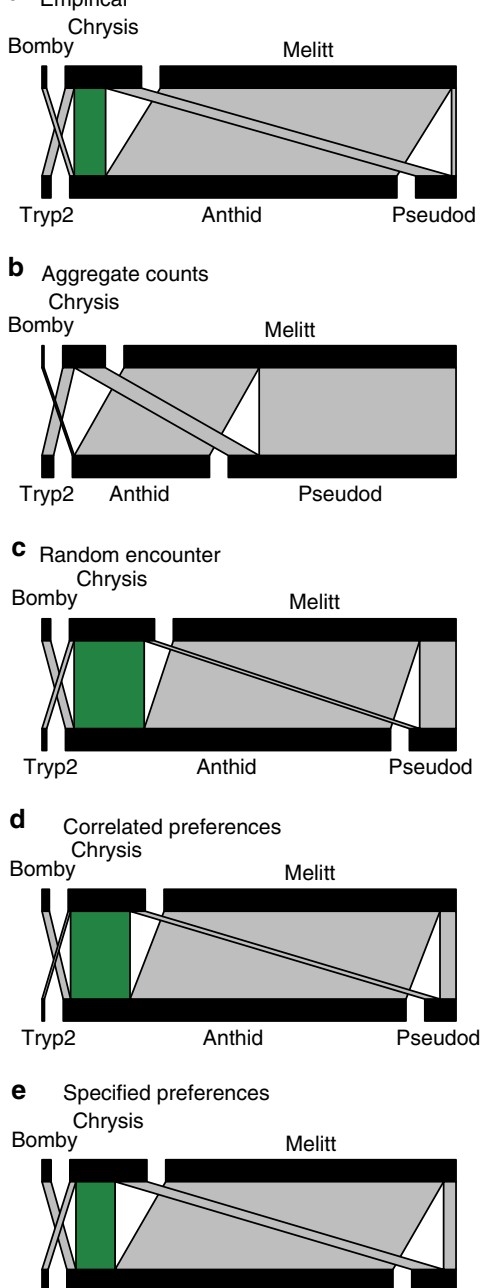

**Fig. 3** Comparison of empirical and predicted weighted network structure at a rice field site in Ecuador. Top-to-bottom: recorded interaction counts (**a**) and predictions of four models (**b**–**e**) that are the same models as in Fig. 2. In each panel, top bars represent parasitoid species (Bomby: Bombyliid Gen. sp.; Chrysis: *Chrysis* sp.; and Melitt: *Melittobia acasta*) and bottom bars represent host species (Anthid: *Anthidium* sp.; Pseudod: *Pseudodynerus* sp.; and Tryp2: *Trypoxylon* sp.2); interaction widths are proportional to the number of recorded or predicted counts, and interactions observed across forest, agroforest and rice habitat types are in grey, while those observed only in the rice habitat type are in green. A field site with relatively few species was chosen for clarity. As in Fig. 2, notice the large improvement in model performance from the aggregate counts model (**b**) to the random encounter model (**c**)

aggregate counts model does not separate changes in relative species abundance from changes in interaction preference, and assumes that recorded interaction frequencies or counts from one habitat type can be used directly to predict weighted network structure at new field sites. Unsurprisingly, the model resulted in poor predictions in modified habitats (Fig. 2), with this poor fit to data clearly evident when examining predicted and observed networks at the level of an individual field site (Fig. 3). This result was expected because, as mentioned above, the relative frequency of interactions is known to change as habitats are modified[10].

Moving to the simple assumption of mass action (i.e., the random encounter model) resulted in more accurate predictions, but unlike with similar habitat types, performance did not improve by using existing preference matrices (i.e., the alternative preferences model; Fig. 4). However, adjusting existing preference matrices based on expected patterns of parasitoid selectivity for hosts in modified habitat types (i.e., the correlated preferences model) substantially improved predictions: Ecuador, $\mathcal{R}_M = 0.43$; Indonesia, $\mathcal{R}_M = 0.4$; Swiss meadow, $\mathcal{R}_M = 0.21$; and Swiss lowland, $\mathcal{R}_M = 0.6$. Interestingly, the correlated preferences model performed least well with the Swiss meadow data set, which comprised only groups with open habitat complexity (but different consumer-resource ratio, see Fig. 1). In turn, the specified preferences model outperformed the correlated preferences model, and with consistently high model performance: Ecuador, $\mathcal{R}_M = 0.87$ with $3/34 = 9\%$ of interaction preferences hardcoded; Indonesia, $\mathcal{R}_M = 0.68$ with $6/35 = 17\%$; Swiss meadow, $\mathcal{R}_M = 0.69$ with $6/38 = 16\%$; and Swiss lowland, $\mathcal{R}_M = 0.65$ with $8/93 = 8\%$. Model performance increased slightly when we combined the specified preferences model with the correlated preferences model (Supplementary Table 3). If the identity of interactions to target and hardcode in the specified preferences model is not known in advance, then a good rule of thumb is to focus on interactions between the more abundant species: Ecuador, $\mathcal{R}_M = 0.87$ with $4/34 = 12\%$ of interaction preferences hardcoded; Indonesia, $\mathcal{R}_M = 0.68$ with $6/35 = 17\%$, Swiss meadow, $\mathcal{R}_M = 0.53$ with $6/38 = 16\%$; and Swiss lowland, $\mathcal{R}_M = 0.39$ with $6/93 = 6\%$ (Supplementary Note 5).

Formal model selection using AIC and BIC[27] favoured the correlated preferences model and the specified preferences model over the other models, including the complete characterisation model (Supplementary Note 5). This result matched our expectation that models that capture systematic changes in interaction preferences provide the most parsimonious combination of model complexity and performance.

## Discussion

A wealth of information about behaviour and species' responses to the environment is contained in weighted interaction networks[2]. However, predictions cannot be made based on empirical networks alone. Ecologists and conservation practitioners need models that combine information from existing networks with other data and theory to make accurate predictions in novel environments. In this study, we compared the performance of seven models and found that simpler models were sufficient to predict network structure at field sites in similar habitat types to existing data, but more complex models were required when field sites were in different habitat types. This result is consistent with our hypothesis that host and parasitoid species behaviour does not change significantly between field sites in similar habitat types but does change significantly between field sites in different habitat types.

Our findings suggest that if network data representative of new field sites are readily available then predicting weighted network structure is straightforward: interaction preferences are likely to

be similar and the alternative preferences model can be used with empirical estimates of species abundance, such as those collected during biodiversity monitoring programmes. For example, the

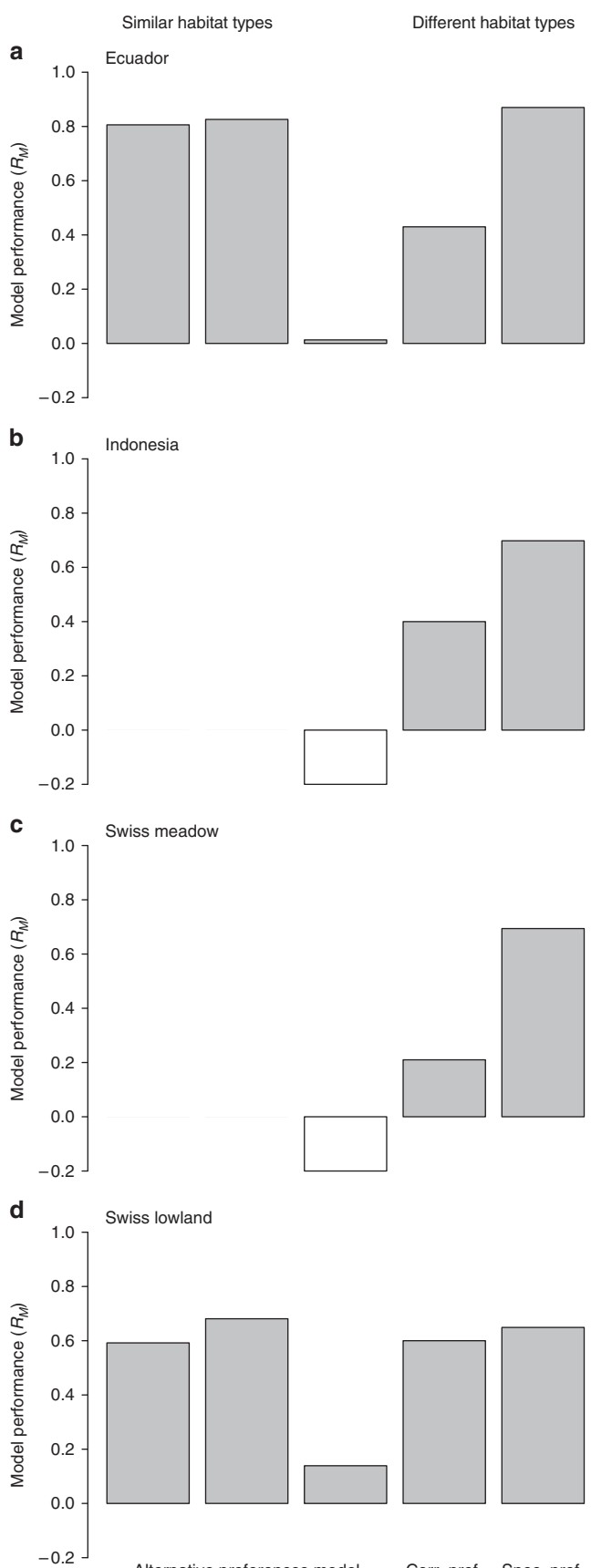

**Fig. 4** Performance of three models based on random species encounter at the group level. Model performance is measured by $\mathcal{R}_M$, which rescales the group-level likelihood of a model to the group-level likelihoods of the random encounter and complete characterisation models (see Eq. 4 in Methods section). Open bars indicate negative values and $\mathcal{R}_M < -0.2$ are capped for display. Left-to-right: predicting between similar habitat types using the alternative preferences model for Ecuador (**a**, pasture-to-rice then rice-to-pasture) and Swiss lowland (**d**, adjacent-to-connected then connected-to-adjacent); then predicting between different habitat types (unmodified-to-modified) using the alternative preferences, correlated preferences and specified preferences models for all four data sets. Indonesia (**b**) and Swiss meadow (**c**) data sets did not contain sufficient interaction data to test predictions between similar habitat types. Notice the high values of $\mathcal{R}_M$ for the alternative preferences model when predicting between similar habitat types, but the low values when predicting between different habitat types. Also notice the improvement in $\mathcal{R}_M$ when using the correlated preferences and specified preferences models for predicting weighted network structure in modified habitat types

interaction preferences inferred here for rice and pasture habitat types could be used to make predictions at new but similar field sites in Ecuador. Of course, it must be recognised that interaction preferences can only be determined if pairs of species have been observed co-occurring already, which may be a limiting factor for predicting weighted network structure in systems with frequent spatial and temporal turnover of community composition. Prediction is more difficult if new field sites are in modified habitat types with limited existing data to inform models, as is the case with most urban habitat types like parks and community gardens. Interaction preferences are likely to be different, and accurate prediction requires understanding which ecological processes and mechanisms are driving these differences. But, as our results for the correlated preferences model show, consistent changes in species behaviour can be mapped to systematic changes in interaction preferences, with measurable benefits for prediction. In addition, the specified preferences model highlights how targeted data collection of particular species and interactions can make predicting the effects of habitat modification more efficient. And given that our models span a range of data requirements, it is possible to customise the trade-off between prediction accuracy and sampling effort depending on the practical question of interest.

In this study, independent measurements of relative species abundance were not available and so predictions were based on effective abundances estimated from network data. By design, our method for estimating relative species abundances will tend to favour an explanation of interaction frequencies in terms of mass action, potentially at the expense of under-estimating genuinely strong or weak interaction preferences. In this regard, it is a conservative method that could under-attribute changes in network structure to species behaviour. It is not currently known how effective abundances correspond to more direct measurements or estimates of species abundance in the field. Although our general approach to prediction is valid either way, determining how effective abundances relate to more direct measurements will be necessary to ensure accurate predictions of weighted network structure. Finding clear relationships between inferred and measured species abundances would also bring about time and cost savings, as only interaction data or abundance data would need to be collected, as appropriate. Identifying such relationships will help with the practical side of prediction, but other kinds of data are needed to clarify the role of species behaviour in determining network structure. This is because our current definition of interaction preferences does not separate 'inherent' preferences from complicating factors due to

the local environment. By 'inherent' preferences, we mean some kind of baseline expectation for how often, for example, a parasitoid would select a particular host given a choice of alternative hosts from different species, but described at the population level rather than the more usual individual level. These 'inherent' preferences are best measured in the controlled setting of laboratory experiments, and doing so would also help untangle the issue of potential and realised niche (Supplementary Note 2). Once measurements have been made, it will then be possible to test more nuanced hypotheses, such as whether 'inherent' preferences are masked in forested habitat types but revealed in open habitat types.

We used a likelihood function based on the multinomial distribution to calculate model performance. This probability distribution is useful because it directly compares model predictions for multiple species to a recorded set of interaction counts. It does so by representing the probability that a parasitoid picks a given host, conditioned on information about other hosts in the community. This conditioning is necessary if, for example, the abundances of particular host species lead to parasitoids forming search images[28] that affect their per capita probabilities of attacking other hosts in the community. As such, the multinomial distribution relies on species richness and community composition being relatively stable over the time period of data collection. Alternatively, one could use a likelihood function based on the binomial distribution, which represents the probability of recording a successful parasitism event given a host-parasitoid encounter in the field, independent of community composition (we discuss other possible probability distributions for the likelihood function in Supplementary Note 2). The binomial distribution assumes that network structure is primarily a pairwise phenomenon, whereas the multinomial distribution assumes that it is primarily a community phenomenon, and likely it is a mixture of the two.

In future work, it will be useful to compile general patterns of shifting interaction preferences between different habitat types, and, indeed, patterns that arise from other forms of environmental change. For example, interaction data could be collected along an altitudinal gradient as a proxy for temperature change, using differences between sets of inferred interaction preferences as the basis of predictive models for climate warming. Identifying which interactions need to be characterised and hardcoded in models is also important for prediction; and the fact that some interactions deviate so strongly from mass action suggests that they may be worth deeper investigation in their own right. Promisingly, we found that only a small fraction of interactions may need to be sampled in modified habitats to significantly improve predictions of network structure, and these interactions likely involve common species with many interaction partners. It will be interesting to apply our models, based on host-parasitoid networks, to other classes of weighted interaction network, such as plant-pollinator networks (in which weights represent the number of recorded visits between species). Although many biological details will of course vary between network classes, separating relative species abundance from other factors affecting network structure will still be useful because our general approach, at its core, represents a fundamental modelling step that is now taken for granted in population dynamical models[29].

With our new methods and models, we can now begin to predict how human-driven change could impact species' interactions in novel environments and unfamiliar conditions. By separating abundance and behaviour, we are better able to compare the functional roles of rare and specialist species to the roles of more abundant and generalist species in a community, both in terms of ecosystem service output and also their relative contributions to network persistence and stability[30–33]. Our approach is also relevant as the final step in a more ambitious

sequence of predictions. Species distribution and demographic models use environmental variables and species' vital rates (e.g., survival, growth, and reproduction) to predict the geographical distribution and abundance of species[34, 35]. The models we have presented can convert these abundances into weighted interaction networks. In this way, we can begin to predict the composition and structure of communities, and, therefore, start assessing and predicting the effects of environmental changes on the global provision of ecosystem services.

## Methods

**Networks and data sets.** We analysed four data sets of weighted networks that describe interactions between insects at two trophic levels: parasitoid species (predators or consumers) and their host species (prey or resources), including information on the number of successful parasitism events (counts) between each host and parasitoid species at the level of a single field site. Mathematically, we represented networks as matrices with entries $B_{ijk}$ that record the number of counts between host species $i$ and parasitoid species $j$ at field site $k$.

The Ecuador data set[10] includes 48 networks sampled from five habitat types: forest (6 networks); shade-grown coffee agroforest (12); abandoned coffee agroforest (6); pasture (12); and rice (12). The Indonesia data set[11] includes 24 networks all sampled from agroforests, and we categorised field sites into two habitat types: more forested (12 networks) and less forested (12). The Swiss meadow data set[12] includes 47 networks sampled from two habitat types: restored meadow (ecological compensation areas, ECAs, 13 networks); and intensively managed meadows at distances 25 m (11), 50 m (12) and 100 m (11) from the nearest ECA. The Swiss lowland data set[13] includes 30 networks sampled from three habitat types: adjacent to forest (10 networks); located at a distance of 100–200 m from the nearest forest but connected by woody elements (10); and isolated at least 100 m away from any woody habitat (10).

We grouped networks by habitat type and determined 12 groups as having sufficient data for analysis. We used metadata to identify two features with each group: habitat complexity (forested or open) and consumer-resource ratio (low or high). Forested-low: Ecuador {forest, coffee, abandoned coffee}; and Indonesia {more forested}. Forested-high: Swiss lowland {adjacent}, {connected}, and {10 most forested from adjacent and connected}. Open-low: Indonesia {less forested}; and Swiss meadow {25 m, 50 m, 100 m}. Open-high: Ecuador {pasture}, {rice} and {pasture, rice}; Swiss lowland {isolated}; and Swiss meadow {ECA}.

**Estimating relative species abundances from interaction data.** Network data often do not include independent measurements of species abundance or local population density, but given sufficient count data it is possible to estimate relative species abundances that are consistent with mass action[24]. These estimates may differ from other, independent measurements of abundance because they represent idealised abundances that provide the closest agreement to data under the mass action hypothesis; they should therefore be considered effective or functional species abundances.

A general form of mass action is $B_{ijk} \propto x_i^{\alpha} x_j^{\beta}$; where $B_{ijk} > 0$ and $x_i$ and $x_j$ are the abundances or local population densities of interacting host and parasitoid species, respectively, and $\alpha$ and $\beta$ are scaling parameters. Notice that this expression for $B_{ijk}$ assumes that abundances hold across a set of field sites indexed by $k$, e.g., all field sites of the same habitat type or in the same group. This is a necessary assumption because there are often insufficient data in individual networks to determine nontrivial estimates of relative species abundance at individual field sites.

Taking logarithms, $\ln(B_{ijk}) \propto \alpha \ln(x_i) + \beta \ln(x_j)$. For a given pair of $\alpha$ and $\beta$ values, if the network ($k = 1$) or group of networks ($k > 1$) is sufficiently dense with interactions then we have a set of over-determined equations[24], with one equation for each recorded $B_{ijk}$. We used the function *lsei* in the R package limSolve[36] to solve this set of equations and obtain estimates of $x_i$ and $x_j$. In practice, we trialled combinations of $0 < \alpha \le 2$ and $0 < \beta \le 2$ in increments of 0.05 and recorded the log-likelihood with $p_{ijk} = f(\alpha, \beta) = \frac{x_i x_j}{\sum_{ij} x_i x_j}$ in Eq. 2. The combination resulting in the largest log-likelihood is the maximum likelihood estimate pair, $\hat{\alpha}$ and $\hat{\beta}$, and we denote the associated maximum likelihood estimate of species abundances by $\hat{x}_i$ and $\hat{x}_j$. As $\alpha$ only controls the distribution of estimated abundances among host species (and similarly with $\beta$ for parasitoid species), our estimates of relative species abundance—our effective abundances—are simply $\hat{x}_i$ and $\hat{x}_j$ (i.e., the expected number of counts for an interaction that follows mass action is proportional to $\hat{x}_i \hat{x}_j$).

**Models.** We developed a series of models for predicting weighted network structure at new field sites in a novel environment. We assessed the performance of models using pairs of groups from the same data set: models were parameterised using data from a calibration group and predictions were tested using recorded counts from a test group, which represents the novel environment. Let us denote variables in the calibration group by $B_{ijk}^{\text{cal}}$, $\hat{x}_i^{\text{cal}}$, $\hat{x}_j^{\text{cal}}$ and $\gamma_{ij}^{\text{cal}}$; and variables in the test group by $B'_{ijk}$, $\hat{x}'_i$, $\hat{x}'_j$ and $\gamma'_{ij}$. Here, we extended the original method[24] for deriving

preference matrices ($\gamma$) from network data to treat interaction data sampled at multiple field sites (Supplementary Note 2). Each model generates probabilities $p_{ijk}$ that are compared to $B'_{ijk}$ using Eq. 2, below, to calculate log-likelihoods; with log-likelihoods then used to measure and compare model performance at individual field sites (Eq. 3) and at the group level (Eq. 4).

Null model with uniform interaction frequencies. All interactions have the same probability, $p_{ijk} = \frac{1}{\sum_{ijk} a_{ijk}}$; where $a_{ijk} = 1$ if $B'_{ijk} > 0$, and zero otherwise, i.e., $\sum_{ijk} a_{ijk}$ is the number of non-forbidden interactions recorded at a field site (ignoring counts).

Aggregate counts model. Probabilities are set proportional to the number of recorded counts summed across networks from different field sites in the calibration group: $p_{ijk} = \frac{\sum_k B_{ijk}^{\text{cal}}}{\sum_{ijk} B_{ijk}^{\text{cal}}}$.

Random encounter model. Probabilities are set proportional to the product of effective abundances of host and parasitoid species in the novel environment: $p_{ijk} = \frac{\hat{x}'_i \hat{x}'_j}{\sum_{ij} \hat{x}'_i \hat{x}'_j}$; recall that effective abundances are assumed to hold across all field sites in a group, which is why there is no $k$-index on the right-hand side of the expression for $p_{ijk}$.

Alternative preferences model. Probabilities are set proportional to the product of an existing preference matrix from the calibration group $\gamma_{ij}^{\text{alt}} = \gamma_{ij}^{\text{cal}}$ and effective abundances in the novel environment: $p_{ijk} = \frac{\gamma_{ij}^{\text{alt}} \hat{x}'_i \hat{x}'_j}{\sum_{ij} \gamma_{ij}^{\text{alt}} \hat{x}'_i \hat{x}'_j}$. Note that the preference matrix from the calibration group is derived using the effective abundances from data in the calibration group (Supplementary Note 2). For switches (interactions known to be possible but with no entry in $\gamma_{ij}^{\text{alt}}$), we considered two possibilities: (i) switches follow mass action and we set $\gamma_{ij}^{\text{alt}} = 1$; or (ii) switches are inherently less-preferred interactions and we set $\gamma_{ij}^{\text{alt}} = 1 - 2^{-\frac{1}{\hat{x}_i^{\text{cal}} \hat{x}_j^{\text{cal}}}}$, which returns values between zero and one, in inverse proportion to the product of effective abundances in the calibration group. As mentioned in the main text, mass action switches consistently led to better model performance, so we present those results only (but see Supplementary Table 2 and Supplementary Note 3).

Correlated preferences model. First, we obtain the column-wise rank order of interaction preferences in $\gamma'_{ij}$, i.e., host species are sorted and identified (first, second, third etc.) from highest-to-lowest interaction preference for each parasitoid species. This rank order represents a systematic pattern in interaction preferences that is identifiable with the novel environment (see Supplementary Note 4). We then reorder entries in $\gamma_{ij}^{\text{cal}}$ (including mass action switches) according to the rank order in $\gamma'_{ij}$ to obtain a new preference matrix: $\gamma_{ij}^{\text{corr}}$. Probabilities are set as $p_{ijk} = \frac{(\gamma_{ij}^{\text{corr}})^{\hat{\delta}} \hat{x}'_i \hat{x}'_j}{\sum_{ij} (\gamma_{ij}^{\text{corr}})^{\hat{\delta}} \hat{x}'_i \hat{x}'_j}$; where $\hat{\delta}$ is a scaling parameter that is applied to each entry in the preference matrix and is set to its maximum likelihood estimate value (we also present results for the model without the optimisation step—that is, with $\hat{\delta} = 1$—in Supplementary Table 3).

Specified preferences model. First, we determine the contribution of each interaction to log-likelihood by calculating Eq. 2 with $B'_{ijk}$ and $p_{ijk} = \frac{\gamma'_{ij} \hat{x}'_i \hat{x}'_j}{\sum_{ij} \gamma'_{ij} \hat{x}'_i \hat{x}'_j}$ with all non-zero entries in $\gamma'_{ij}$ set to one except the focal entry. We sort the log-likelihood contributions and identify the interactions with any obvious discontinuity (see Supplementary Fig. 2). We then replace—hardcode—the entries for these influential interactions in $\gamma_{ij}^{\text{cal}}$ (including mass action switches) with their corresponding values in $\gamma'_{ij}$ to obtain a new preference matrix: $\gamma_{ij}^{\text{spec}}$. Probabilities are set as $p_{ijk} = \frac{\gamma_{ij}^{\text{spec}} \hat{x}'_i \hat{x}'_j}{\sum_{ij} \gamma_{ij}^{\text{spec}} \hat{x}'_i \hat{x}'_j}$. The specified preferences and correlated preferences models can be combined by hardcoding entries for the influential interactions in $\gamma_{ij}^{\text{corr}}$ (see above) rather than $\gamma_{ij}^{\text{cal}}$.

Complete characterisation model. All interaction preferences must be characterised individually in the novel environment and so the relevant preference matrix is $\gamma_{ij}^{\text{complete}} = \gamma'_{ij}$. Probabilities are set as $p_{ijk} = \frac{\gamma_{ij}^{\text{complete}} \hat{x}'_i \hat{x}'_j}{\sum_{ij} \gamma_{ij}^{\text{complete}} \hat{x}'_i \hat{x}'_j}$. The model results in the best fit to data possible in our current approach and, by definition, returns the maximum model performance at the group level. It is worth emphasising that the model does not result in perfect fit to data, which would correspond to log-likelihood equal to zero; rather, the log-likelihood at the group level ($\mathcal{L}_{\text{complete}}$ in Eq. 4) indicates how well a single preference matrix is able to explain weighted network structure at multiple field sites in the same group.

**Likelihood function for testing model fit**. We assumed that the number of recorded counts, $B_{ijk} > 0$, between host species $i$ and parasitoid species $j$ at field site $k$ follows a multinomial distribution[20]. The corresponding likelihood function for a set of recorded counts generated with probabilities $p_{ijk}$ is

$$L(p_{ijk}|B_{ijk}) = P(B_{ijk}|p_{ijk}) = \frac{(\sum_{ijk} B_{ijk})!}{\prod_i \prod_j \prod_k B_{ijk}!} \prod_i \prod_j \prod_k (p_{ijk})^{B_{ijk}} \quad (2)$$

and the log-likelihood is $\mathcal{L} = \ln(L)$, which we calculated using the function dmultinomin in R[36].

**Model performance at individual field sites**. We measured the performance of model $M$ at field site $k$ as

$$\mathcal{F}_{M,k} = \frac{\mathcal{L}_{\text{null},k} - \mathcal{L}_{M,k}}{\mathcal{L}_{\text{null},k}} \quad (3)$$

where the null model is described above and $\mathcal{L}_{\text{null},k}$ and $\mathcal{L}_{M,k}$ are log-likelihoods calculated using Eq. 2 with a single $k$-index. $\mathcal{F}_{M,k} = 1$ if model $M$ completely explains the distribution of recorded interaction counts at field site $k$; $\mathcal{F}_{M,k} = 0$ if it performs the same as the null model; and $\mathcal{F}_{M,k} < 0$ if it performs worse than the null model.

**Model performance at the group level**. We measured the performance of model $M$ at the group level as

$$\mathcal{R}_M = \frac{\mathcal{L}_{\text{re}} - \mathcal{L}_M}{\mathcal{L}_{\text{re}} - \mathcal{L}_{\text{complete}}} \quad (4)$$

where the random encounter (re) and the complete characterisation (complete) models are described above, and $\mathcal{L}_{\text{re}}$, $\mathcal{L}_{\text{complete}}$ and $\mathcal{L}_M$ are log-likelihoods calculated using Eq. 2 for all field sites in a group of networks together, and, therefore, with multiple $k$-indices. $\mathcal{R}_M = 1$ if model $M$ performs as well as the complete characterisation model; $\mathcal{R}_M = 0$ if it performs the same as the random encounter model; and $\mathcal{R}_M < 0$ if it performs worse than the random encounter model.

**Code availability**. Computer code can be accessed by contacting the corresponding author (P.P.A.S.).

**Data availability**. Host-parasitoid networks can be accessed by contacting the appropriate author (Ecuador: O.T.L. or J.M.T.; Indonesia: A.M.K.; Swiss meadow: M.A.; Swiss lowland: V.C.).

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

## Acknowledgements

We thank Céline Bellard, Georgina Mace and Daniel Stouffer for comments, and Matt Walters for producing Fig. 2. P.P.A.S. was supported by an AXA Postdoctoral Research Fellowship and a Postdoctoral Fellowship from the National Socio-Environmental Synthesis Center (SESYNC) funded by National Science Foundation DBI-1052875, O.T.L. by NERC grant NE/N010221/1, J.M.T. by a Rutherford Discovery Fellowship administered by the Royal Society of New Zealand, M.A. by European commission grant QLRT-2001-01495 and Swiss Federal Office for Science and Technology grant 01.0524-2, V.C. by British Ecological Society grant 4785/5824 awarded to P.P.A.S., A.-M.K. by German Science Foundation grant DFG: KL 1849/5-2 and F.R.-T. by James Martin 21st Century Foundation grant LC1213-006.

## Author contributions

P.P.A.S. was responsible for research planning, analysis and writing; O.T.L., J.M.T. and F.R.-T. for additional research planning and writing; and M.A., V.C., A.-M.K., O.T.L. and J.M.T. provided data. All authors discussed the results and edited the manuscript.

## Additional information

**Competing interests:** The authors declare no competing financial interests.

