## [Peer review file · Nature Communications]

Reviewers' comments:

Reviewer #2 (Remarks to the Author):

In this manuscript the authors test a number of different statistical models for the outcome of Habitat modification. The main methodological innovation is to use models that separate species abundances from interaction preferences. Perhaps the most exiting result of this exercise is that detailed observation of certain pairs of species can be informative for the wider system.

The methodology of the paper is very simple and and the methodological advance is incremental. The authors use very basic mathematics and the separation of abundance and interaction preference follows basic modelling intuition. While this might be new in the context of statistical models of habitat modification this separation exists in every (!) population dynamical model. The authors obtain some nice topical results, but these are not of the broad importance that I would expect in a Nature Comms paper.

The authors use existing data sets that are then evaluated using the different, but very basic, models under consideration. In terms of substance this appears to be very much on the lighter side. In particular I find it worrying that conclusions are drawn from a small set of numbers without any challenging test or statistical serious statistical testing. I realize that may be partly due to the nature of the question, on which data is very limited, but still the wider readership of nature Comms would find this one hard to swallow.

In summary, this is good work, but it presents an incremental methodical advance and does not present topical results that are of sufficient substance or importance to merit publication in Nature Communications.

Reviewer #3 (Remarks to the Author):

This paper is a very smart approach to further developments in ecological network analysis, especially about its predictability.

Separating abundance effects and interaction preferences, and considering prey switching are interesting and great developments. Methodology is clearly presented and not over-complicated. The paper is clear and well-written.

One issue is the taxonomic (not temporal) aggregation of networks. Several ecological processes are probably best characterizable at the level of functional groups, not species. Even if we do have high-resolution data, these may not be the best (most relevant). A brief discussion on whether and why to describe these communities at the species level would add to the paper. Clearly, both the questions and the answers are different in more aggregated networks, the question is how to best describe the key ecological processes.

Since we only have host-parasitoid networks (great databases), one obvious question emerges: what do the Authors think about the generality of the results? Would results be similar to pollinators? Any intuitive idea about some differences?

One question is how much we can know about the real preferences of species, based on the raw

measurements even in a totally undisturbed system? Would be nice to read a little bit more about the potential/realized niche issue here. Since probabilities are combined in the different models, it is crucial to have a good assessment on the basic preferences (e.g. like 3-species subsystems in isolation).

One technical idea. Figure 4 may be slightly changed: (1) the columns are too wide, this is not nice and (2) the location should be given above the chart and the model name should be given under the chart.

There is a mass of information in the supplementary material. This is all right but, at least, 2.7 should be discussed briefly in the main text, I think. This is so crucial (the width of the sampling time window and temporal aggregation, see <http://www.sciencedirect.com/science/article/pii/S0304380009003238>).

A: the Authors compare different model scenarios for predicting the effects of habitat changes on host-parasitoid networks.

B: very interesting and original approach.

C: excellent database

D: methodology is powerful but simple. AIC and BIC well used.

E: conclusions are solid but not oversold.

F: both taxonomic and temporal aggregation better discussed.

G: one paper suggested to consult, otherwise all right.

H: enjoyable read, easy to understand.

Reviewer #4 (Remarks to the Author):

Staniczenko et al. Predicting the effect of habitat modification on networks of interacting species

Submitted to Nature Communications

This manuscript reports an analysis of the effect of habitat modification on pairwise species interactions. Results are compiled at the community level, allowing an interpretation of the effect of habitat modification on network structure. I previously evaluated this manuscript for another journal and gave a favorable review. I maintain my position, I find this study interesting, stimulating and innovative. It could be a relevant contribution for Nature Communications because it improves our understanding of why ecological networks vary from one place to another. Traditionally, community ecology compares lists of species across sites; now we see a paradigm shift with the comparison of ecological networks. Such a representation of a community is much more inclusive because it not only represents the species that are present and changing from a location to another, but also the way they interact. The second contribution of this manuscript is technical, it brings new models to represent the interactions and the methods that are developed could be useful for other studies. I must admit I do similar stuff myself and I am still puzzled/challenged by some decisions that were taken to represent interactions in their statistical model.

That said, the presentation of the study is way too technical for a general audience journal such as the one of Nature Communications. The authors report the results of 'models', not the underlying ecology. Such a presentation would be ok if the object of the paper was to develop new methods and if the authors were targeting a more specialized audience, but this is not the case. The way the objective of the study is formulated is just one good example. At P2L46 it is said that the object of the study is to 'test whether we can accurately predict the effects of habitat modification on the structure of weighted host-parasitoid networks'. Further, a sub-objective is to evaluate the amount and type of data that has to be included in the models to perform appropriate predictions. A more general and encompassing objective would be to partition the drivers of network turnover with habitat modification. Predicting networks (actually what is done is predicting pairwise interactions across an entire set of species) is a technical objective, understanding the drivers of network turnover is a more conceptual objective. The structure of the manuscript follows this perspective, with subtitles corresponding to each of the model. Results are reported by type of model, which makes the story way too complicated. Basically, I would summarize the results in just a few paragraphs. A first one ranking the hypotheses (not models) for the 'between habitat types with similar levels of modification' and the 'between habitat types with different levels of modification'. What is important is the relative support for each model, not so much their individual performance.

The description of the methods, both in the main text and in the specific Methods section tends to be confusing. It has to be edited to become more accessible for a general audience. This is second time I evaluate this manuscript, and I do this kind of analysis myself, and even then I had to read it twice to make sure I do understand. I recognize that some research is simply too complicated and requires efforts to understand, but it is not the kind of papers this journal is looking for.

As an example, the terminology used to describe the hierarchical nature of the data is very difficult to follow and rather inconsistent. If I try to summarize, there are four datasets, corresponding to four studies, conducted at different regions, at different field sites, each with different habitat types (sometimes referred as, environment or levels of habitat modification), consumer-resource ratio and groups. The groups are defined as 'regional scale properties' (L72), which is in itself confusing because multiple regions are lumped in each group. The consumer-resource ratio is not defined (it is the ratio of what that is computed?). I am not contesting the classification, but it looks like the terms used to refer to these different descriptors of the data are not used consistently. Sometimes with synonyms. For instance, P19L409 in the methods, abundances are compiled across a set of field sites - there is not enough information for me to understand how abundance was compiled. I guess this is within a group? Within a group for datasets coming from a given region/study/dataset? This is just a simple example of a problem that spans the entire manuscript. It makes it particularly hard to understand the model comparisons. I understand the nature of the data can't be simplified really, but the way results are reported could be.

In the text and the methods, I think it would help making the story more accessible if the models were first described by their ecology, and then with the maths. As written right now, I have to go through all equations and interpret them, which was difficult with with a priori knowledge of the study and of similar approaches. I can't believe it would be accessible to someone interested by the topic but without the appropriate qualifications in modelling.

There are still some hidden assumptions in the model formulation that are unclear to me. I do get why the mass action assumption is used to estimate relative abundance, but I find it bizarre that this assumption is also made latter in the 'random encounter model', and indirectly involved as well in the other models where p_{ijk} is computed. It "looks" problematic, but perhaps additional explanation could help solving this issue. First, it sounds like the data is used twice to compute the likelihood, giving a false impression of circularity. The interaction count data feeds a first model, used to estimate relative abundance, and then as the data in the likelihood function (which also includes the p_{ijk} , using the presently estimated abundances). The second problem, which is more

problematic conceptually, is that the neutrality assumption is made to estimate abundance, and then relative abundance is used in non-neutral models (in the correlated, specified and complete models). Obviously, if these models are right, then the estimate of relative abundance is wrong. The only solution I see (there might be others) is to estimate everything in a single step, via a hierarchical model.

The other technical point that is not explained enough is the conceptual justification for the multinomial distribution. I remember I've made this point before and I am sorry to see no change in response to it. It could be a binomial, or a Poisson distribution. The multinomial implicitly assumes there is a fixed number of trials and that there is a tradeoff between potential hosts when a parasitoid is searching for them (ie a total number of trials, spread across the different species). A binomial model would treat all species pairs independently and provide exactly the same likelihood whatever is the composition of the community. In other words, the decision of taking a multinomial has for consequences that the number of counts between two pairs of species will change with the change in the composition of the community from one location to another, even if the two species are exactly the same absolute abundance. I am not convinced of this as a general fact (it might be ok though for parasitoids that do have a single interaction in their life), but I recognize it is an intuition and that research will have to be conducted to compare the two approaches. At least, the authors need to detail why this model, and anticipate the consequences of their decisions relative to other options. Not that I am in favour of one or another, I think research still has to be conducted to determine which is the right distribution, but the justification has to be provided.

Further, to me abundance has to come somewhere in the distribution used in the likelihood function. The number of trials for the multinomial process has to depend on the abundance of the two species, not the counts. My view of the problem is rather different and I would like to exchange with the authors about it, or at least provide better justifications for their approach. I rather see a hierarchical model where you have the expected number of encounters:
$$N_{ijk} \propto x_{ik}^a * x_{jk}^b$$

With the the interaction probability:
$$p_{ijk} = \gamma_{ijk} / \sum_{ij} \gamma_{ijk}$$

And the number of interactions resulting from a binomial process $P(B_{ijk} | p_{ijk}, N_{ijk})$. The random encounter model would still be possible, but for the case with γ equal across all species. As a consequence, I don't see conceptually the difference between the aggregate counts model and the random encounter model. The bottom line of my comment at the end is that the model has to be grounded on the ecology, not just described passively as it is right now. This type of approach is highly innovative and therefore it needs to be better justified.

There is one information I missed from the methods: what if a species pair in the 'test group' is absent from the 'calibration group' ? And the other way around ?

I haven't understood how is γ is computed in the novel environment (L463) ?

'Predictive capability'- I don't have the same interpretation. To me it is simply a different null model from the one used to compute explanatory power (albeit some changes in the formulation of the equation).

I signed my evaluation
Dominique Gravel

Response to reviewer comments

This document accompanies a revised manuscript with all changes highlighted (rewritten text in orange and added text in magenta); also included is a version without highlighting that is more suitable for printing. Below, reviewer comments are quoted in full in typewriter font, and each point is followed by our response. Page and line references in our responses correspond to the revised manuscript, unless stated otherwise. Please note that there is no Reviewer 1 because one of the original referees was unfortunately unable to deliver a report and another referee was therefore recruited in his/her place.

Reviewer 2

In this manuscript the authors test a number of different statistical models for the outcome of Habitat modification. The main methodological innovation is to use models that separate species abundances from interaction preferences. Perhaps the most exiting result of this exercise is that detailed observation of certain pairs of species can be informative for the wider system.

We thank the reviewer for his/her careful reading of our manuscript.

The methodology of the paper is very simple and and the methodological advance is incremental. The authors use very basic mathematics and the separation of abundance and interaction preference follows basic modelling intuition. While this might be new in the context of statistical models of habitat modification this separation exists in every (!) population dynamical model. The authors obtain some nice topical results, but these are not of the broad importance that I would expect in a Nature Comms paper.

Although our methodological contributions may seem simple from the perspective of population dynamical models, we suggest that they are not just an incremental contribution but are truly novel, as noted by the other two reviewers. Empirical studies have only begun to document how environmental changes affect complex ecological networks over the past 10 years, so already being able to predict these impacts is a huge advance. We expect that our approach will be used by a broad range of ecologists and biologists, both theoreticians and empiricists, as well as researchers working with complex networks more generally, e.g., Saavedra, Reed-Tsochas and Uzzi (2009) A simple model of bipartite cooperation for ecological and organizational networks, *Nature* 457, pp. 463–466.

It is true that almost all population dynamical models inherently separate species abundance from interaction preferences. Surprisingly though, almost all work on ecological networks does not make this fundamental separation when analysing network structure, particularly when considering the effects of environmental change, e.g., Tylanakis, Tschamntke and Lewis (2007) Habitat modification alters the structure of tropical host-parasitoid food webs, *Nature* 445, pp. 202–205. Indeed, one motivation for this work was to introduce to field and theoretical ecologists a simple yet powerful approach to separating the effects of abundance from those of interaction preferences when dealing with network data. Our approach is also intended to be useful when independent measurements of species abundance are not available, which is the

case with most studies involving ecological networks.

We acknowledge the reviewer's point in our discussion (P19L401): "Although many biological details will of course vary between network classes, separating relative species abundance from other factors affecting network structure will still be useful because our general approach, at its core, represents a fundamental modelling step that has been taken for granted in population dynamical models for decades [Brauer and Castillo-Chavez 2000]."

The authors use existing data sets that are then evaluated using the different, but very basic, models under consideration. In terms of substance this appears to be very much on the lighter side. In particular I find it worrying that conclusions are drawn from a small set of numbers without any challenging test or statistical serious statistical testing. I realize that may be partly due to the nature of the question, on which data is very limited, but still the wider readership of nature Comms would find this one hard to swallow.

The reviewer is correct with his/her impression that there is very limited data with which to validate the effects of environmental change on ecological networks. Moreover, we were very careful when choosing which data sets to analyse to ensure that our results were comparable and robust (a point noted by Reviewer 3, who highlighted that we had assembled an "excellent database"). As mentioned in the previous version of the Supplementary Information (footnote on P15), we set a very high bar for a data set to be included in our study. Specifically, we required that a data set satisfy three criteria for inclusion: i) the lower trophic level involves cavity-nesting insects, so that the method of data collection was similar among data sets; ii) networks were sampled from what could broadly be considered "modified" and "unmodified" habitats, so that we could assess the effect of habitat modification on weighted network structure; and iii) more than ten networks were sampled from sites in each different type of habitat, so that we could assess changes in weighted network structure both at the level of individual field sites and at a larger geographical scale (i.e., across multiple field sites at once).

To add further context, even among the comprehensive collection of antagonistic interaction networks in Morris, Gripenberg, Lewis and Roslin (2014) Antagonistic interaction networks are structured independently of latitude and host guild, *Ecology Letters* 17, pp. 340–349, very few of the data sets satisfy our three criteria. Of 28 data sets, only 3 satisfy all three criteria (these 3 data sets are included in our study). Only 6 data sets involve cavity-nesting insects; only 5 data sets include networks sampled from both "modified" and "unmodified" habitats; and only 8 data sets include ten or more replicate networks in total (i.e., irrespective of whether networks were sampled in different habitat types), with 14 data sets containing only a single network (see Table S1.1 in the Supplementary Information of Morris et al.). We also analysed a fourth suitable data set from a more recent study: Coudrain, Schüepp, Herzog, Albrecht and Entling (2014) Habitat amount modulates the effect of patch isolation on host-parasitoid interactions, *Frontiers in Environmental Science* 2, pp. 27. On the topic of data set suitability, it is worth emphasising that all four data sets analysed were collected by authors of the present study. We acknowledge that the lack of suitable data has prevented rigorous statistical testing across data sets, but are optimistic that the data requirements and modelling ideas advanced in this study will inform and direct future data collection.

In summary, this is good work, but it presents an incremental methodical advance and does not present topical results that are of sufficient substance or importance to merit publication in Nature Communications.

We thank the reviewer for his/her critical but balanced assessment of our work. We hope that our responses, along with the improved presentation in the revised manuscript following suggestions by Reviewer 4, more clearly highlights the novel methodological contributions, conceptual advances and practical usefulness of our new approach to predicting the structure of weighted ecological networks.

Reviewer 3

This paper is a very smart approach to further developments in ecological network analysis, especially about its predictability.

We thank the reviewer for taking the time to consider our work and are pleased with his/her positive assessment.

Separating abundance effects and interaction preferences, and considering prey switching are interesting and great developments. Methodology is clearly presented and not over-complicated. The paper is clear and well-written.

We find it reassuring that the reviewer was able to follow the methodology in the original manuscript. Nevertheless, given the comments of the other two reviewers, we felt it was necessary to improve the presentation in the revised manuscript so that the widest possible audience could understand and implement our new approach. We hope that the reviewer still considers the revised manuscript to be clear and well-written.

One issue is the taxonomic (not temporal) aggregation of networks. Several ecological processes are probably best characterizable at the level of functional groups, not species. Even if we do have high-resolution data, these may not be the best (most relevant). A brief discussion on whether and why to describe these communities at the species level would add to the paper. Clearly, both the questions and the answers are different in more aggregated networks, the question is how to best describe the key ecological processes.

This is a good point that has both theoretical and practical implications. Due to space constraints, we have added a paragraph about the taxonomic aggregation of networks in the Supplementary Information (Section 1.4) and direct the reader to this discussion on P8L160. The new paragraph reads: "Our approach is based on food webs resolved to the level of individual species. At least in host-parasitoid systems, this is likely to be the most relevant level of analysis because of the intimate, co-evolved relationships between hosts and parasitoids, and their relatively high specificity. This is also the level of aggregation most frequently used in empirical network studies, although separation within species (e.g., according to genotypes) or aggregations of species are occasionally published. More importantly, species-level partitioning of host resources across parasitoids has been shown empirically to relate to functional properties such as attack rates and their stability [Peralta et al. Ecology 2014], so there is

an established relationship between network architecture with species as nodes and ecosystem processes. For larger community networks, particularly those that are more taxonomically and functionally diverse [Montoya et al. Nature Communications 2015], and which include a wider range of interactions types (e.g., predator-prey food webs), a definition of nodes in terms of aggregations of taxa may be equally or more illuminating about dynamics compared to an approach focused on species-level interactions. Nevertheless, the general approach and methods introduced in this study are applicable to taxonomically aggregated networks, but care must be taken that predictive models reflect ecological mechanisms and processes that are appropriate for the level of aggregation under investigation.”

Since we only have host-parasitoid networks (great databases), one obvious question emerges: what do the Authors think about the generality of the results? Would results be similar to pollinators? Any intuitive idea about some differences?

We thank the reviewer for pressing us to answer these important questions and have extended our discussion (P19L399): “It will be interesting to apply our models to other classes of weighted interaction network, such as plant-pollinator networks (in which weights represent the number of recorded visits between species). Although many biological details will of course vary between network classes, separating relative species abundance from other factors affecting network structure will still be useful because our general approach, at its core, represents a fundamental modelling step that has been taken for granted in population dynamical models for decades.”

For the benefit of the reviewer, we would like to share some initial thoughts from early analyses of two other data sets that involve different species from those in host-parasitoid networks. As analysis is ongoing, it would not be appropriate to include such preliminary findings in the current manuscript. First, we are applying our approach to an unpublished (at the time of writing) data set of 360 plant-pollinator networks sampled in urban environments. Interaction data were collected as part of a single study across four large cities and nine different urban habitat types. Initial findings suggest that the “alternative preferences model” can be used to successfully predict network structure between similar habitat types (e.g., private gardens and allotments), but not dissimilar habitat types (e.g., private gardens and industrial estates). We have yet to explore which and in what way more complex models (e.g., the “correlated preferences model”) may be useful for making predictions about urban plant-pollinator communities.

Second, we are analysing a published data set of four mosquito-host networks (two sampled indoors and two sampled outdoors) where, unusually, information is also provided on the relative abundance of host and mosquito species. We find that our approach using preference matrices can be successfully combined with independent measurements of species abundance to predict network structure at new locations. Although tested using a comparatively small data set, we find that our approach outperforms a more conventional approach based on forage (or sometimes “foraging”) ratios—see Hess, Hayes and Tempelis (1968) The use of the forage ratio technique in mosquito host preference studies, *Mosquito News* 28, pp. 386–389—that we have adapted for use with interaction networks. Interestingly, the performance of the “alternative preferences model” can be improved by systematically adjusting host abundance distributions derived from survey data. Specifically, the largest gain in model fit results from reducing the effective proportion of humans compared to farm animals when making predictions at outdoor locations. We suggest this is because survey data lead to overestimates of outdoor encounter

rates between mosquitoes and their hosts.

One question is how much we can know about the real preferences of species, based on the raw measurements even in a totally undisturbed system? Would be nice to read a little bit more about the potential/realized niche issue here. Since probabilities are combined in the different models, it is crucial to have a good assessment on the basic preferences (e.g. like 3-species subsystems in isolation).

This is an interesting question. Due to space constraints, we have added a paragraph about the issue of potential and realised niche in the Supplementary Information (Section 2.7) and direct the reader to this discussion on P19L396. The new paragraph reads: “In the main text, we suggested that studying single-parasitoid-multiple-host subsystems under controlled conditions would help untangle the issue of potential and realised niche. Although in this study we have focused on predicting the effects of habitat modification on the feeding preferences of species, it is worth emphasising that even in undisturbed systems we cannot be certain of measuring what could be considered underlying or ‘true’ preferences. This is because the feeding interactions observed in these systems may be a subset of those that could be achieved in the absence of competitors: we are likely to be observing the realised feeding niches of these species, and their fundamental feeding niches may be broader (or differ in magnitude) from those observed under natural conditions. The only way to fully resolve the relationship between potential and realised niche would be a series of manipulative experiments where preferences are assessed across all sets of hosts for each parasitoid species, in isolation from competition. This is not feasible in most systems. But in the absence of data on true preferences, although imperfect, our version of preferences are likely to be at least broadly accurate and may be useful as a first step at predicting changes to weighted network structure.”

One technical idea. Figure 4 may be slightly changed: (1) the columns are too wide, this is not nice and (2) the location should be given above the chart and the model name should be given under the chart.

We have made the suggested changes to Figure 4 (P34).

There is a mass of information in the supplementary material. This is all right but, at least, 2.7 should be discussed briefly in the main text, I think. This is so crucial (the width of the sampling time window and temporal aggregation, see <http://www.sciencedirect.com/science/article/pii/S030438009003238>).

We have moved and adapted the discussion about aggregating networks that was previously in Section 2.7 of the Supplementary Information to the main text, which also includes the suggested citation (P7L146): “It is worth emphasising the importance of modelling species abundances at the level of individual field sites, even if it is necessary to assume the same value for effective abundance at multiple field sites. This is because representing system properties using spatially aggregated data can give misleading results. For example, consider five networks that each contain the same two host ($i = 1$ and $i = 2$) and parasitoid species ($j = 1$). When aggregated, there are 15 counts to the first host and 10 counts to the second host; and we therefore estimate relative host species abundance as $X_{i=1} = 15$ and $X_{i=2} = 10$. However, following our suggested approach, we find that effective abundances are $\hat{x}_{j=1} = 1$ for the para-

sitoid species, and $\hat{x}_{i=1} = 3$ and $\hat{x}_{i=2} = 6$ for the two host species. In this example, the relative magnitudes of host species abundance at the level of individual field sites is the reverse of the estimate based on aggregated data. This is because, on closer inspection, we might find that the number of recorded counts at the five field sites is something like $\{3, 3, 3, 3, 3\}$ for the first host and $\{10, 0, 0, 0, 0\}$ for the second host, which means that the second host ($i = 2$) should really be modelled as being more abundant than the first host ($i = 1$). In general, using the sum of counts across networks as a proxy for abundance or population density will underestimate values for spatially less-common species. Similar issues arise with temporal data aggregation [Jordán and Osváthc Ecological Modelling 2009]. We discuss the related topic of aggregating networks by species taxonomy in *Supplementary Information*.”

A: the Authors compare different model scenarios for predicting the effects of habitat changes on host-parasitoid networks.

B: very interesting and original approach.

C: excellent database

D: methodology is powerful but simple. AIC and BIC well used.

E: conclusions are solid but not overselled.

F: both taxonomic and temporal aggregation better discussed.

G: one paper suggested to consult, otherwise all right.

H: enjoyable read, easy to understand.

Again, we thank the reviewer for thoroughly assessing our work.

Reviewer 4

Staniczenko et al. Predicting the effect of habitat modification on networks of interacting species

Submitted to Nature Communications

This manuscript reports an analysis of the effect of habitat modification on pairwise species interactions. Results are compiled at the community level, allowing an interpretation of the effect of habitat modification on network structure. I previously evaluated this manuscript for another journal and gave a favorable review. I maintain my position, I find this study interesting, stimulating and innovative. It could be a relevant contribution for Nature Communications because it improves our understanding of why ecological networks vary from one place to another. Traditionnally, community ecology compares lists of species across sites; now we see a paradigm shift with he comparison of ecological networks. Such a representation of a community is much more inclusive because it not only represents the species that are present and changing from a location to another, but also the way they interact. The second contribution of this manuscript is technical, it brings new models to represent the interactions and the methods that are developped could be useful for other studies. I must admit I do similar stuff myself and I am still puzzled/challenged by some decisions that were taken to represent interactions in their statistical model.

We thank the reviewer (Professor Gravel) for assessing our work for a second time and continuing to offer a favourable review with constructive feedback. We have implemented all of his suggestions, which we believe has significantly improved the manuscript. Furthermore, we hope that our detailed responses to some of the reviewer's more specific comments, below, will help clarify and justify our statistical choice for representing interactions when determining model likelihoods.

That said, the presentation of the study is way too technical for a general audience journal such as the one of Nature Communications. The authors report the results of 'models', not the underlying ecology. Such a presentation would be ok if the object of the paper was to develop new methods and if the authors were targetting a more specialized audience, but this is not the case. The way the objective of the study is formulated is just one good example. At P2L46 it is said that the object of the study is to 'test whether we can accurately predict the effects of habitat modification on the structure of weighted host-parasitoid networks'. Further, a sub-objective is to evaluate the amount and type of data that has to be included in the models to perform appropriate predictions. A more general and encompassing objective would be to partition the drivers of network turnover with habitat modification. Predicting networks (actually what is done is predicting pairwise interactions across an entire set of species) is a technical objective, understanding the drivers of network turnover is a more conceptual objective. The structure of the manuscript follows this perspective, with subtitles corresponding to each of the model. Results are reported by type of model, which makes the story way too complicated. Basically, I would summarize the results in just a few paragraphs. A first one ranking the hypotheses (not models) for the 'between habitat types with similar levels of modification' and the 'between habitat types with different levels of modification'. What is important is the relative support for each model, not so much their individual performance.

We agree that the presentation of the manuscript was far too technical for a general audience. We have rewritten large parts of the text following the reviewer's excellent suggestions. In general, his advice to put the ecology first and technical details second has, we feel, greatly improved the readability and accessibility of our work. There is too much rewritten text to directly quote in this response, but they are highlighted in orange in the revised manuscript. In some cases, we have added text to the manuscript, which are highlighted in magenta.

In the introduction, we provide clearer ecological motivation for separating relative species abundance and interaction preferences (P2L37). We also state the hypothesis (P3L62) that we are testing in ecological terms and how this relates to the modelling (with a brief summary of results and conclusions on P3L69) and return to the original hypothesis in light of our findings in the discussion (P17L359). Our attempts at improving the accessibility of our methods and models are described in detail, below, in response to other queries raised by the reviewer.

The description of the methods, both in the main text and in the specific Methods section tends to be confusing. It has to be edited to become more accessible for a general audience. This is second time I evaluate this manuscript, and I do this kind of analysis myself, and even then I had to read it twice to make sure I do understand. I recognize that some research is simply too complicated and requires efforts to understand, but it is not the kind of papers this journal is looking for.

We agree. In addition to rewriting large parts of the main text, we have expanded the Methods section to more clearly describe our models, such that a general reader would be able to implement them in his/her own studies.

As an example, the terminology used to describe the hierarchical nature of the data is very difficult to follow and rather inconsistent. If I try to summarize, there are four datasets, corresponding to four studies, conducted at different regions, at different field sites, each with different habitat types (sometimes referred as, environment or levels of habitat modification), consumer-resource ratio and groups. The groups are defined as 'regional scale properties' (L72), which is in itself confusing because multiple regions are lumped in each group. The consumer-resource ratio is not defined (it is the ratio of what that is computed?). I am not contesting the classification, but it looks like the terms used to refer to these different descriptors of the data are not used consistently. Sometimes with synonyms. For instance, P19L409 in the methods, abundances are compiled across a set of field sites - there is not enough information for me to understand how abundance was compiled. I guess this is within a group? Within a group for datasets coming from a given region/study/dataset? This is just a simple example of a problem that spans the entire manuscript. It makes it particularly hard to understand the model comparisons. I understand the nature of the data can't be simplified really, but the way results are reported could be.

We thank the reviewer for highlighting specific parts of the text that were difficult to understand and have tried to clarify all points of confusion in the revised manuscript. We have improved the description of the hierarchical nature of the data (P5L94): “We analysed each study as a separate data set and organised data into a three-level hierarchy: network, group of networks, and data set. Each network was built from interaction data collected at a single field site. For mathematical convenience, we represent weighted networks as matrices with entries B_{ijk} that record the number of interactions, also referred to as counts, between host species i and parasitoid species j at field site k . To test hypotheses more easily, we grouped networks by habitat type and used metadata to identify two features with each group: habitat complexity and consumer-resource ratio.”

We have reduced the amount of jargon and number of synonyms, and have been much more consistent with the use of each term. For example, we have cut all reference to “regional-scale properties [of networks],” which in hindsight just added confusion. We now only use “habitat type” and no longer refer to “levels of habitat modification” and “environment” in reference to our data sets.

We have added a definition of the consumer-resource ratio to the main text (P5L102): “We defined consumer-resource ratio as the total number of parasitoids (successful parasitism events across all species) divided by the total number of parasitised and unparasitised hosts collected in the field. This measure indicates how easily parasitoids are able to locate their hosts in particular habitat types, and we labelled groups as being associated with either “high” or “low” consumer-resource ratio (see *Supplementary Information*).”

We have clarified the relationship between recorded counts, interaction preferences and effective abundances. First, we clarify the scope of models and comparisons in our approach (P6L134): “Th[e] decomposition [$B_{ijk} \propto \gamma_{ij}\hat{x}_i\hat{x}_j$] assumes that a single preference matrix is valid across all field sites included in the set of k -indices under consideration, e.g., across all networks in the same group. This assumption is useful for prediction because a single, model-generated preference matrix can then be used to determine weighted network structure at multiple field sites.” Then we provide additional details about effective abundances (P7L138): “We refer to \hat{x}_i and \hat{x}_j as effective abundances because they can be considered a functional property of the system that contributes directly to recorded interaction counts, and also to make it clear that their value may be different from other estimates of species abundance, such as those obtained from survey data. As explained in *Methods*, the above decomposition assumes that effective abundances hold across all field sites in the same habitat type. This is a necessary assumption because there are often insufficient data in individual networks to determine non-trivial estimates of relative species abundance at individual field sites, and explains why there is no k -index attached to effective abundances despite it being mathematically more desirable.”

In the text and the methods, I think it would help making the story more accessible if the models were first described by their ecology, and then with the maths. As written right now, I have to go through all equations and interpret them, which was difficult with with a priori knowledge of the study and of similar approaches. I can't believe it would be accessible to someone interested by the topic but without the appropriate qualifications in modelling.

This is an excellent suggestion that we have taken to heart and tried to follow when rewriting the main text and methods. When explaining how interaction preferences can be used to make predictions, we provide some ecological reasoning for models (P9L191): “The most complex model [in our approach] assumes that changes in species behaviour are so complex that each interaction preference in the matrix must be characterised individually when making predictions at a new field site—we call this the *complete characterisation model*. An intermediate model, which we call the *alternative preferences model*, assumes that species behaviour changes very little between habitat types, and so the preference matrix derived from one group of networks is useful for predicting weighted network structure at a new field site. This model is expected to perform well at making predictions between similar habitat types, but poorly if the new field site is in a very different habitat type from the one used for calibration.”

We provide additional ecological context when describing the aggregate counts model (P11L229): “Existing analyses implicitly assume that recorded counts have intrinsic predictive value, such that interaction data from one habitat type can be used to make predictions at field sites in other habitat types without additional data processing. To illustrate the benefit of separating changes in relative species abundance from changes in interaction preference, we present results for the *aggregate counts model*, which does not make this separation.” Also

for the random encounter model (P11L239): “Mechanism-based approaches to weighted networks typically assume that interaction frequencies are a function of relative species abundance [Vázquez et al. Oikos 2007]. The simplest application of this idea in our approach is the random encounter model, which assumes no species behaviour and so all non-zero entries in the corresponding preference matrix have the same value.”

We provide practical motivation for the specified preferences model (P14L303): “Of course, we could improve model performance by empirically characterising all changes in interaction preference in modified habitat types, but such exhaustive data collection is costly and time consuming. It would therefore be useful to know how many and what kind of interactions should be targeted for empirical study.” We have also added a practical benefit of the model (P16L323): “For future predictions, influential interactions should be studied in experiments and targeted when sampling data at new field sites. If the identity of interactions to target is not known in advance, then a good rule of thumb is to focus on interactions between the more abundant species. [...] Because abundant species are more likely to be detected in any sample, our results suggest that low-intensity samples could be augmented by the specified preferences model to yield reliable quantification of network structure at substantially lower cost.”

There are still some hidden assumptions in the model formulation that are unclear to me. I do get why the mass action assumption is used to estimate relative abundance, but I find it bizarre that this assumption is also made later in the ‘random encounter model’, and indirectly involved as well in the other models where p_{ijk} is computed. It ‘‘looks’’ problematic, but perhaps additional explanation could help solving this issue. First, it sounds like the data is used twice to compute the likelihood, giving a false impression of circularity. The interaction count data feeds a first model, used to estimate relative abundance, and then as the data in the likelihood function (which also includes the p_{ijk} , using the presently estimated abundances). The second problem, which is more problematic conceptually, is that the neutrality assumption is made to estimate abundance, and then relative abundance is used in non-neutral models (in the correlated, specified and complete models). Obviously, if these models are right, then the estimate of relative abundance is wrong. The only solution I see (there might be others) is to estimate everything in a single step, via a hierarchical model.

This comment made it very clear that we needed to do a much better job of explaining and justifying our methodology for estimating relative species abundances (we refer to these estimates as “effective abundances”), and how this step fits into the workflow for making predictions. We have updated the revised manuscript accordingly, most notably by adding a five-step outline of how we use effective abundances and interaction preferences to make predictions (P9L181): “Testing this approach using our data sets involved five steps. First, we selected a calibration and test group from the same data set. Second, we inferred effective abundances from interaction data in the test group to represent values at the new field site. Third, we used a predictive model to generate a preference matrix based primarily on information from the calibration group. Fourth, we combined the effective abundances with the preference matrix to produce a predicted set of interaction counts (Eqn 1). And fifth, we assessed model performance by comparing the predicted distribution of counts among species to the recorded distribution in

the test group. These steps were then repeated for another pair of calibration and test groups.”

We hope that the above text addition and the other rewrites in the revised manuscript has clarified matters and will prevent future confusion for readers. Below we include an additional discussion for the reviewer that goes into more detail about his specific concerns. We begin with a summary of our treatment of species abundances then directly address the two issues raised by the reviewer.

Estimating species abundances

The starting point of any mechanistic approach to modelling the relative frequency of species interactions is relative species abundance. Unfortunately, independent measurements of species abundance rarely accompany data about ecological networks. This means that we must first obtain estimates of relative species abundance before embarking on more complicated modelling involving species behaviour etc. As such, our methodological approach involves two distinct parts: first, we estimate relative species abundance from interaction data; then second, we use those newly estimated species abundances in our models for predicting weighted network structure. Previously, we failed to adequately separate the two parts in our presentation of the methods, which was confusing for the reader.

In the first part, we obtain estimates of relative species abundance at the group level, i.e., using interaction data from multiple networks with each network built from data collected a single field site. For example, imagine that we have 10 networks for forest habitat type in Ecuador. We use interaction data from all 10 networks to estimate one set of relative species abundances that we assume is representative of forest habitat in Ecuador. When estimating relative species abundances, we assume two scaling parameters, α and β , that allow us to find the most neutral distribution of species abundances. As α and β are “free” parameters, we use interaction data to find their maximum likelihood estimate (mle) values $\hat{\alpha}$ and $\hat{\beta}$. It is important to emphasise that finding mle values for α and β is a very different procedure from calculating the likelihood of our models given observed network data (the focus of the second part, discussed below), and the two steps should be performed separately from one another. When computing mle values for α and β , we are using data to inform the most conservative estimates of relative species abundance under the assumption of neutrality, with mass action as the only process determining interaction frequency; but when computing likelihoods in the second part, we are comparing how well the outputs of our models can explain empirical networks. The result of the first part is a vector of relative species abundances \hat{x} (what we call “effective abundances”), containing entries \hat{x}_i for hosts and \hat{x}_j for parasitoids, that we use as inputs in models for predicting network structure.

In the second part, we assume that the vector of relative species abundances \hat{x} is an objective property of the system under investigation (in this example, a representative distribution of species abundances for forest habitat type in Ecuador). These \hat{x} are used in our models, the simplest of which is the “random encounter model,” which assumes that empirical interaction frequencies can be adequately explained by the product of host and parasitoid abundances, i.e., $\sim \hat{x}_i \hat{x}_j$. Our more complicated models then modify this baseline expectation using the concept of interaction preferences. As discussed in more detail below, the output of these models for network structure can be compared to empirical data using a number of different likelihood functions (we use the likelihood function associated with the multinomial distribution).

Issue 1: Data used twice to compute likelihood

The reviewer's first issue is the concern that data is used twice to compute the likelihood of a model, giving the false impression of circularity. Although we hope that we have shown above that data is not used twice to compute the likelihood, there remain a few subtle points related to this issue that are worth discussing for complete transparency. The first part can be thought of as an optimisation step where we find the mle values of two parameters (α and β) that result in the most conservative estimate of a neutral distribution of species abundances, whereas the second part uses likelihood as a way of comparing predictive models to empirical data (and therefore predictive models to one another). As such, when we present likelihoods for the "random encounter model" we are presenting the best possible results for this model. This is because by using mle values ($\hat{\alpha}$ and $\hat{\beta}$) we are calculating the best likelihoods that can be attributed to the assumption of random encounter, out of all possible models that are consistent with random encounter (i.e., among all possible combinations of α and β). This is actually an advantage of our approach, as more complicated models must explain more of the empirical data to be judged notably better than the simple "random encounter model" (for reference, we also present results for the "random encounter model" with $\alpha = 1$ and $\beta = 1$ in the Supporting Information, which by definition results in worse likelihoods than the corresponding model with $\alpha = \hat{\alpha}$ and $\beta = \hat{\beta}$). Because all of our more complicated models use the same vector of relative species abundances \hat{x} , we can fairly judge to what extent incorporating additional ecological processes into models improves predictions.

Issue 2: Validity of neutrality assumption about estimated species abundances

The reviewer's second issue involves the validity of our neutrality assumption about estimated species abundances. As mentioned above, we suggest that our approach to estimating relative species abundances under the assumption of neutrality results in the most conservative values possible given available data. However, it is worth noting that if a "genuine" interaction preference is very strong then some of this preference may be absorbed into our abundance estimates of the interacting species. In other words, in some instances we may be inadvertently attributing to greater relative abundance what is, in reality, a greater preference for a particular interaction (i.e., our estimated value for the interaction preference would be lower than the "true" value). So yes: given that we must *estimate* relative species abundance, there will be some discrepancy between our estimated values ("effective abundances") and the "genuine" values. Discrepancies will be small when the system is well-explained by mass action/random encounter and potentially larger when specialist parasitoids are involved in interactions with very strong, "genuine" interaction preference. This is because when a parasitoid species is involved in few interactions it is more parsimonious to attribute a relatively large number of interaction counts to high abundance and mass action rather than to an especially strong interaction preference. However, we suggest that our approach is reasonable given the type of data typically available, and look forward to conducting a formal assessment of our approach to estimating relative species abundance once independent measurements are collected along with interaction data.

The other technical point that is not explained enough is the conceptual justification for the multinomial distribution. I remember I've made this point before and I am sorry to see no change in response to it. It could be a binomial, or a Poisson distribution. The multinomial implicitly assumes there is a fixed number of trials and that there is a tradeoff between potential hosts when a parasitoid is searching for them (ie a total number of trials, spread across the different species). A binomial model would treat all species pairs independently and provide exactly the same likelihood whatever is the composition of the community. In other words, the decision of taking a multinomial has for consequences that the number of counts between two pairs of species will change with the change in the composition of the community from one location to another, even if the two species are exactly the same absolute abundance. I am not convinced of this as a general fact (it might be ok though for parasitoids that do have a single interaction in their life), but I recognize it is an intuition and that research will have to be conducted to compare the two approaches. At least, the authors need to detail why this model, and anticipate the consequences of their decisions relative to other options. Not that I am in favour of one or another, I think research still has to be conducted to determine which is the right distribution, but the justification has to be provided.

Further, to me abundance has to come somewhere in the distribution used in the likelihood function. The number of trials for the multinomial process has to depend on the abundance of the two species, not the counts. My view of the problem is rather different and I would like to exchange with the authors about it, or at least provide better justifications for their approach. I rather see a hierarchical model where you have the expected number of encounters: $N_{ijk} \propto x_{ik}^a * x_{jk}^b$

With the the interaction probability:
 $p_{ijk} = \gamma_{ijk} / \sum_{ij} \gamma_{ijk}$

And the number of interactions resulting from a binomial process $P(B_{ijk} | p_{ijk}, N_{ijk})$. The random encounter model would still be possible, but for the case with gamma equal across all species. As a consequence, I don't see conceptually the difference between the aggregate counts model and the random encounter model. The bottom line of my comment at the end is that the model has to be grounded on the ecology, not just described passively as it is right now. This type of approach is highly innovative and therefore it needs to be better justified.

This is a very interesting line of thinking. Motivated by the reviewer's comment, we thought further about the various probability distributions that could be used to calculate likelihoods for our models and data. In the end, we believe the multinomial distribution is the most appropriate probability distribution for our models and currently available data, but an approach using the binomial distribution—as suggested by the reviewer—is potentially useful given additional data (specifically, independent measurements of relative species abundance). We intend to compare the two distributions in future work, so thank the reviewer for raising this thoughtful point.

We have added our motivation for using the multinomial distribution likelihood function in the main text (P10L208): “We quantified the accuracy of model predictions using a likelihood function based on the multinomial distribution [Vázquez et al. Ecology 2009] (see *Methods*, Eqn 2). We chose this likelihood function because it describes how well a model is able to explain the recorded distribution of interaction counts among species at a field site.”

Below, we justify our use of the multinomial distribution for computing likelihoods and discuss the application of six alternative probability distributions:

- i. Binomial
- ii. Poisson binomial
- iii. Poisson
- iv. Compound Poisson
- v. Hypergeometric
- vi. Multivariate hypergeometric

First, we describe our general approach to modelling weighted interaction networks. Then we describe our use of the multinomial distribution for computing likelihoods. This is followed by an in-depth discussion of a potential approach involving the binomial distribution. Finally, we briefly cover the remaining five probability distributions.

Modelling approach

Consider for simplicity an empirical, weighted bipartite network that we represent by the matrix \underline{B} with interaction counts B_{ij} (recorded successful parasitism events) that are indexed by host species i and parasitoid species j . Let the total number of counts in the matrix be $F = \sum_{ij} B_{ij}$.

Given matrix \underline{B} , we first estimate a vector of best-fit relative species abundances \hat{x} , containing entries \hat{x}_i for hosts and \hat{x}_j for parasitoids (as discussed above). From each of our models, we obtain a matrix of interaction preferences $\underline{\gamma}$ (with entries γ_{ij}) that is designed to modify the expected frequency of interactions according to mass action to provide a more accurate prediction of interaction counts (e.g., with the random encounter model $\gamma_{ij} = 1$ for all i and j , for non-forbidden interactions).

In order to compare the output of a model to network data using likelihood, we obtain the relative probability of each interaction given the observed species composition of the local community, i.e., we are interested in how well a model explains the relative distribution of counts in an empirical matrix. The appropriate probability for an interaction for use in the multinomial distribution is $p_{ij} = \frac{\gamma_{ij}\hat{x}_i\hat{x}_j}{\sum_{ij}\gamma_{ij}\hat{x}_i\hat{x}_j}$.

To a certain extent we are assuming that if a parasitoid encounters a host ($\propto \hat{x}_i\hat{x}_j$) then it does not forgo parasitising the host. In other words, we are modelling encounters as events that lead directly to parasitism—and therefore counts in an interaction matrix—which is why we refer to \hat{x} as “effective abundances.” The preference matrix then represents whether there is either (i) some kind of foraging preference ($\gamma_{ij} > 1$) or avoidance $\gamma_{ij} < 1$ compared to well-mixed random encounter (mass action), or (ii) the encounter is unlikely to lead to a successful parasitism event for the parasitoid ($\gamma_{ij} < 1$), e.g., due to the fact that only one parasitoid can emerge from an individual host even though multiple parasitoids may have infected the host. As discussed below, this approach can be thought of as condensing an approach using the binomial

distribution with two steps (encounter then infection or not) to a single step (biased encounter), which is necessary due to limited data on relative species abundance.

Ideally, we would also have independent abundance data (from a species survey, for example) that could be used to explore how empirical measurements of relative species abundance convert to effective abundances (we make this point in the Discussion section, P18L382). As our effective abundances can be considered the functional property of the system that contributes directly to recorded interaction counts, it would be interesting to see how independent abundances map to effective abundances, and whether this mapping varies by species or even interaction. We are intending to collect this additional data in future work.

Multinomial distribution

Given the focus of our discussion, it is worth emphasising that the multinomial distribution is a generalisation of the binomial distribution. However, we suggest there are subtle but important differences between the two probability distributions when computing likelihoods with network data in terms of ecological interpretation. Specifically, the binomial distribution is useful for explaining the count/frequency of an individual interaction while the multinomial distribution is useful for explaining the distribution of interaction counts/frequencies across a community. We suggest they are complementary approaches and intend to compare their advantages and disadvantages in future work once additional data become available.

In general, the multinomial distribution models the probability of counts when rolling a k -sided die n times. For n independent trials each of which leads to a success for exactly one of k categories, with each category having a fixed success probability, the multinomial distribution gives the probability of any particular combination of successes for the various categories. When $n > 1$ and $k = 2$ the multinomial distribution is the binomial distribution.

As such, the multinomial distribution is useful for describing how interaction counts are distributed among matrix elements. We imagine this is why it was chosen for the likelihood function in Vázquez, Chacoff and Cagnolo (2009) Evaluating multiple determinants of the structure of plant-animal mutualistic networks, *Ecology* 90, pp. 2039–2046 (our inspiration for using the multinomial distribution). The multinomial distribution allows us to ask: given that we have recorded a total of F counts in the field, how are counts distributed among possible interactions according to different predictive models and how do these predictions compare to the empirical distribution of counts?

With our modelling approach, the likelihood function for the multinomial distribution is $L_{\text{multinomial}} = \frac{F!}{\prod_i \prod_j B_{ij}} \prod_i \prod_j p_{ij}^{B_{ij}}$. As $p_{ij} = f(\hat{x}_i, \hat{x}_j)$, see above, our likelihood function does take into account species abundance (or, at least, our version of species abundances in terms of effective abundances). We stress that although the calculation of likelihood is tied to a specific data matrix with a fixed number of species and counts (as, indeed, would be the case with any likelihood calculation), our models still generate an important system-specific property that is applicable to communities with different species composition and therefore matrices of different sizes—this property is the set of preferences γ_{ij} rather than the set of probabilities p_{ij} .

Binomial distribution

As mentioned above, the binomial distribution is a special case of the multinomial distribution and is useful for describing individual interactions rather than multiple interactions. In

our context, the number of trials n is most naturally understood as the number of encounters between a particular pair of species, i.e., $n = n_{ij}$, and the probability that an encounter leads to an interaction count in \underline{B} is given by a distinct probability $q = q_{ij}$. With this formulation, the likelihood function for the binomial distribution is $L_{\text{binomial}} = \frac{n_{ij}!}{B_{ij}!(n_{ij}-B_{ij})!} q_{ij}^{B_{ij}} (1 - q_{ij})^{n_{ij}-B_{ij}}$; where the recorded number of counts B_{ij} is used for the number of “successes.”

A tricky question given the data we have available is how to determine appropriate values for n_{ij} and q_{ij} . Regarding n_{ij} , it is not advisable to use effective abundances \hat{x} for two reasons. First, the product $n_{ij} \sim \hat{x}_i \hat{x}_j$ is not guaranteed to give an integer. Second, an effective abundance is not guaranteed to reflect the general abundance of a species, rather, it represents the component of abundance that directly contributes to counts in an interaction network/matrix. Furthermore, if $n_{ij} = f(\hat{x}_i, \hat{x}_j)$ then it is not clear that it would be reasonable to use $q_{ij} = f(\gamma_{ij})$, so we would need to devise an alternative means of determining the “preference” of an interaction.

We suggest that the binomial distribution is most appropriate when independent measurements of species abundance are available in addition to interaction data. The binomial distribution then offers a nice two-step approach to calculating likelihoods, as suggested by the reviewer. Let us represent independent abundances by \underline{X} , indexed as X_i for hosts and X_j for parasitoids. In Step I, we can use independent abundances to model the expected number of encounters between each pair of species, i.e., $n_{ij} \sim X_i X_j$, which could also be modified by an additional multiplicative factor to reflect differences among interactions due to environmental or foraging effects. In Step II, we can use interaction data to determine the conversion probability for an interaction count given an encounter, i.e., $q_{ij} = f(\gamma_{ij})$ or something similar. The new interpretation of “preference” then represents whether an individual parasitoid “chooses” to parasitise the host following an encounter with a particular host, and/or, following an infection, whether internal competition within the host results in a successful parasitism event or not for the parasitoid. (With plant-pollinator visitation networks, Step II would capture whether a pollinator physically “decides” to land on a plant or not.)

Framed in this way, it becomes clear that our current approach with preferences $\underline{\gamma}$ and the multinomial distribution essentially works by condensing Steps I and II into a single step (which is necessary given the lack of data on independent species abundances).

In summary, the binomial distribution is conceptually more straightforward and arguably better reflects ecological processes at the level of individual interactions, but the multinomial distribution is more appropriate for our network data and has the desirable feature of simultaneously modelling multiple interactions in a community.

Poisson binomial distribution

Having outline a promising approach to calculating likelihoods using the binomial distribution, it is worth noting that there is a second more general form (besides the multinomial distribution) in the Poisson binomial distribution.

The Poisson binomial distribution is the discrete probability distribution of a sum of independent Bernoulli trials that are not necessarily identically distributed. In other words, it is the probability distribution of the number of successes in a sequence of n independent yes/no experiments with success probabilities q_1, q_2, \dots, q_n . The ordinary binomial distribution is the special case when $q_1 = q_2 = \dots = q_n$.

There is a subtle point of difference compared to the binomial distribution when considering

multiple interactions with the Poisson binomial distribution: the likelihood function associated with the Poisson binomial distribution would measure the ability of a model to generate the same number of total counts (i.e., F) in an empirical matrix, rather than the likelihood of each interaction individually.

Poisson distribution

As our data are matrices of counts, we can consider using a Poisson distribution to calculate likelihoods. With our modelling approach, the likelihood function for the Poisson distribution is $L_{\text{Poisson}} = \frac{\lambda_{ij}^{B_{ij}} e^{-\lambda_{ij}}}{B_{ij}!}$; where λ_{ij} is the expected number of counts for the interaction between host species i and parasitoid species j . In using the Poisson distribution, however, we would lose many of the nice links to ecological processes and mechanisms afforded by the multinomial and binomial distributions. The Poisson distribution would be a somewhat crude model for computing the likelihood for this and similar systems, with its application largely motivated by the simple fact that we are dealing with count data.

Compound Poisson distribution

Similar to the relationship between the binomial distribution and the Poisson binomial distribution, the compound Poisson distribution is a potential extension of the Poisson distribution to multiple interactions in a community. The likelihood function for the compound Poisson distribution is $L_{\text{compoundPoisson}} = \prod_{ij} \frac{\lambda_{ij}^{B_{ij}} e^{-\lambda_{ij}}}{B_{ij}!}$. We suggest that for our purposes the compound Poisson distribution suffers from the same disadvantages as the regular Poisson distribution.

Hypergeometric distribution

The hypergeometric distribution is a discrete probability distribution that describes the probability of k successes in n draws, without replacement, from a finite population of size N that contains exactly K successes, wherein each draw is either a success or a failure. By contrast, the binomial distribution describes the probability of k successes in n draws with replacement.

For our system, computing likelihoods using the hypergeometric distribution requires an approximation for the total number of encounters between two species, e.g., $N_{ij} \propto X_i X_j$; and an approximation for the total number of success, e.g., $K_{ij} \propto \gamma_{ij} \hat{x}_i \hat{x}_j$. The likelihood function for the hypergeometric distribution then compares these N_{ij} and K_{ij} to the observed number of successes $k_{ij} = B_{ij}$ given some sampling parameter n_{ij} . Framed in this way, it is clear that using the hypergeometric distribution to compute likelihoods is currently not feasible with available data. It is, however, potentially useful in future to investigate the effect of sampling effort on the structure of weighted interaction networks. For example, once N_{ij} and K_{ij} have been adequately parameterised, one could explore how k_{ij} varies under different n_{ij} .

Multivariate hypergeometric distribution

Similar to the relationship between the binomial distribution and the multinomial distribution, the multivariate hypergeometric distribution is a potential extension of the hypergeometric distribution to multiple interactions in a community. Specifically, the multivariate hypergeometric

distribution is the without-replacement equivalent to the multinomial distribution (which, like the binomial distribution, assumes replacement).

There is one information I missed from the methods: what if a species pair in the 'test group' is absent from the 'calibration group' ? And the other way around ?

We cover this point under the term “switches” and now include more details in the main text (P10L202): “We modelled switches (interactions present in the test group but not calibration group) in two ways: i) switches follow mass action; or ii) switches are inherently less-preferred interactions (see *Methods*). Assuming mass action switches consistently led to better model performance, so we present those results only (it is worth noting, however, that some switches had interaction preferences that differed significantly from one, see Supplementary Table 2).” We provide a thorough discussion about switches in the Supporting Information (Section 3).

To directly answer the reviewer’s questions, if the interaction between host species i and parasitoid species j is not recorded in the “calibration group” but is recorded in the “test group” then we set its value for interaction preference to the most parsimonious value $\gamma_{ij} = 1$, which assumes that the interaction follows mass action (an evaluation of an alternative approach where the interaction is assumed to be unobserved because it is inherently less-preferred is included in the Supporting Information). There is no need to assign a value for the interaction preference if no counts were recorded for a host-parasitoid pair in the “test group” when the multinomial distribution is used to calculate likelihood. If the binomial distribution were used instead, then one could use information from the “calibration group” to see how much a predictive model ($\gamma_{ij}\hat{x}_i\hat{x}_j$) overestimates the observed number of counts—which is zero—in the “test group.”

I haven’t understood how is gamma’ is computed in the novel environment (L463) ?

We have rewritten our description of the “specified preferences model” referred to by the reviewer, both in the main text (P14L300) and Methods section (P23L497), and have extended our description of the model in the Supplementary Information (Section 5.6).

‘Predictive capability’ - I don’t have the same interpretation. To me it is simply a different null model from the one used to compute explanatory power (albeit some changes in the formulation of the equation).

We agree with the reviewer and have removed all reference to “predictive capability.” Instead, we refer to \mathcal{R}_M as a measure of model performance at the group level. We now also refer to $\mathcal{F}_{M,k}$ (Eqn 3 in *Methods*) as model performance at individual field sites, and have improved our motivation for using the two measures (P11L219): “For a given model, we found that $\mathcal{F}_{M,k}$ varied greatly among networks in the same group, which was potentially masking differences in model performance (Supplementary Fig. 2). This variation was due, in part, to our use of a single preference matrix to predict weighted network structure at all field sites in a group (see Eqn 1). So to better compare model performance, we also used the measure \mathcal{R}_M (Eqn 4 in *Methods*), which describes model performance at the group level. This measure still compares predicted to recorded counts at individual field sites, but involves calculating likelihood for all field sites in a group at once. With \mathcal{R}_M , the likelihood of model M is rescaled to the likelihood of the simple random encounter model (corresponding to $\mathcal{R}_M = 0$) and the likelihood of the maximally-complex complete characterisation model (corresponding to $\mathcal{R}_M = 1$).”

Reviewers' comments:

Reviewer #2 (Remarks to the Author):

In the response to my earlier criticism the authors have restated their point, therefore let me restate my previous assessment: The paper presents a nice case study, but all the theory is essentially already well established in population dynamics. I do not judge the results to be of sufficient novelty for publication in Nature Comms.

Reviewer #3 (Remarks to the Author):

The Authors clearly and successfully addressed all of my questions. I do believe that the relatively simple nature of the methods is a merit, not a limitation. The paper is acceptable for publication.

Reviewer #4 (Remarks to the Author):

I will start by giving my appreciation of the authors' commitment to respond to my comments. It was a fruitful exercise, the reply was an interesting discussion and the manuscript is greatly improved.

One of the unfortunate consequences of this generous reply however, is that the manuscript is now way too long and essential results are lost in a flow of descriptions. Despite introducing more ecology into the description of the models (which will be appreciated I'm sure), the manuscript remains technical, as exemplified by the abstract and the structure of the main text. First, we can't find in the abstract the answer to the authors' hypothesis. We can only find the take-home at the end of the first discussion paragraph, where it is stated that differences in preferences among habitat types are more important than differences in relative abundances. The last two sentences of the abstract should be revised to emphasize the findings. This would indirectly please the first reviewer, who was not convinced of the novelty of the results.

But most of all, the report of the results is still provided model by model. I would appreciate a much more synthetic presentation, with ranking of models instead. The result section should make only a few paragraphs, and details description should be provided as supplementary information.

2. I still find a coherence problem with the way abundance is estimated in non-neutral models (e.g. complete model). Obviously, there is a conceptual issue with first estimating abundance assuming interactions are driven by mass-action only, and then introducing species-specific preferences in the likelihood computation. This issue is however not only conceptual, as the approach could bias the estimates and alter the final model comparison. Imagine for instance that a given host species is highly vulnerable because of its traits (e.g. lack of defenses). The estimated relative abundance of this species will be high in the mass-action model and as a consequence, this will lower the estimated gamma values in the complete model, much below the 'reality' (it will basically tend to 1). It poses obviously a problem for the interpretation of the gammas, but also it could bias the inference in the altered habitat. Imagine for instance that abundance of that species is reduced considerably in the altered habitat. The predicted interaction probabilities, across all parasitoids, will consequently decrease much more than they should if the estimated gamma were much larger than 1. As a result, the fit of the 'complete model' in the altered habitat will appear much worse than it should if the relative abundance was appropriately estimated.

The bottom line is that the abundance should be estimated simultaneously as the preference to prevent biases.

3. I appreciate the efforts to compare the different distribution and a summary of this discussion should find its place somewhere in the methods. But I must say I remain skeptical of the multinomial model. The argument that the multinomial is a generalization of the binomial distribution is irrelevant in the actual context, because ecologically speaking they describe two totally different stochastic processes. On the one hand, the binomial model represents the probability that an interaction occurs if you present a host to a given adult parasitoid, whatever is the community composition. This stochastic process describes for instance the probability that an egg laid in the host will develop into an adult, which has nothing to do with community composition. On the other hand, the multinomial model represents the probability that the parasitoid picks a given host, knowing all other potential hosts. The multinomial model has the advantage of accounting for higher order interactions (e.g. if there is active foraging), but this could also be accounted for in a conditional binomial model. Further, there should be a multinomial model for each community composition.

It remains to be proven, I agree, but as I said my intuition is that the interaction between two species is a pairwise phenomenon, not a community one. Any individual parasitoid is not sampling the entire community before setting into a host. Likely, the adult will attempt oviposition on the first host it finds. The development of the larvae will depend on the quality of the host, irrespective of the composition of the community. Or may be, with enough information, we would find it is a mixture of the two.

A numerical example will better illustrate my point. Consider a community with host abundances $x_i = \{1,6,3\}$ and relative preferences $\gamma = \{1,2,0.5\}$. The interaction probability with the first host will be $1x_1 / (1x_1 + 6x_2 + 3x_3 \cdot 0.5) = 1/14.5 = 0.069$. If for whatever reason, the third host species is absent at another location, this interaction probability will change to $1x_1 / (1x_1 + 6x_2) = 1/13 = 0.077$. This might be reasonable if the total number of larvae remains the same and we have to spread them among a reduced set of hosts. This is unlikely to be the reality, but it could be a reasonable assumption for host-parasite interactions. I am far from convinced however that the same reasoning would apply to other types of interactions where the number of plant visits or killing events by predators is not constant.

Pushed to the extreme, the problem with the multinomial model is that the interaction probability between a given pair of host and parasitoid will converge to 1 as other species are removed from the community. The interaction then becomes deterministic.

All of this raises concerns in the interpretation of what is exactly the stochastic process the model is describing. My fear is that actually, the two processes above described (ie the pairwise interaction and the selection function) are mixed. Perhaps the authors are clear about this, but it is not in the main text and will obviously not be for the reader that has never thought about interactions as a probabilistic process.

To be more constructive, I would suggest the authors to better describe what is the stochastic process they represent, and consequently why they choose a multinomial model. Finally, the hidden assumptions of this decision should be clearly explained.

Response to reviewer comments

This document accompanies a revised manuscript with all changes highlighted (rewritten text in orange and added text in magenta); also included is a version without highlighting that is more suitable for printing. Below, reviewer comments are quoted in full in typewriter font, and each point is followed by our response. Page and line references in our responses correspond to the revised manuscript, unless stated otherwise. Please note that there is no Reviewer 1 because one of the original referees was unfortunately unable to deliver a report and another referee was therefore recruited in his/her place.

Reviewer 2

In the response to my earlier criticism the authors have restated their point, therefore let me restate my previous assessment: The paper presents a nice case study, but all the theory is essentially already well established in population dynamics. I do not judge the results to be of sufficient novelty for publication in Nature Comms.

We thank the reviewer for continuing to consider our manuscript. We would like to stress that we are not attempting to re-invent the theory of population dynamics (as we make clear in the main text, P17L355). We have continued to revise the manuscript in accordance with Reviewer 4's suggestions to better highlight the "novelty of the results."

Reviewer 3

The Authors clearly and successfully addressed all of my questions. I do believe that the relatively simple nature of the methods is a merit, not a limitation. The paper is acceptable for publication.

We thank the reviewer for continuing to consider our manuscript and are pleased that he or she believes it is acceptable for publication.

Reviewer 4

I will start by giving my appreciation of the authors' commitment to respond to my comments. It was a fruitful exercise, the reply was an interesting discussion and the manuscript is greatly improved.

We are pleased that the reviewer finds the manuscript greatly improved and that he appreciated our previous reply. We also thank him for continuing to provide constructive feedback. Indeed: If the manuscript is deemed suitable for publication, then we would like to thank the reviewer (who signed his review) by name in the acknowledgements section.

One of the unfortunate consequences of this generous reply however, is that the manuscript is now way too long and essential results are lost in a flow of descriptions. Despite introducing more ecology into the description of the models (which will be appreciated I'm sure), the manuscript remains technical, as exemplified by the abstract and the structure of the main text. First, we can't find in the abstract the answer to the authors' hypothesis. We can only find the take-home at the end of the first discussion paragraph, where it is stated that differences in preferences among habitat types are more important than differences in relative abundances. The last two sentences of the abstract should be revised to emphasize the findings. This would indirectly please the first reviewer, who was not convinced of the novelty of the results.

We have added two sentences to the abstract that answers our hypothesis and clarifies our take-home message (P1L24): "The models map to ecological mechanisms and we find that interaction preferences change significantly between different habitat types but not between similar habitat types. This difference likely reflects changes in species behaviour, and models that capture systematic changes in interaction preferences provide the best predictive ability relative to their data requirements."

But most of all, the report of the results is still provided model by model. I would appreciate a much more synthetic presentation, with ranking of models instead. The result section should make only a few paragraphs, and details description should be provided as supplementary information.

We agree. We have completely revised the presentation of models in the results section, which we feel makes the manuscript much more accessible and practically useful. This rewriting has shortened the manuscript by over 500 words. We thank the reviewer for insisting that we make this change. We now provide a short, ecologically motivated description of the models in two paragraphs (P9L191), and have separated model results into two subsections: "Predicting between similar habitat types" (P12L242) and "Predicting between different habitat types" (P2L251). Model results are introduced in order of complexity and data requirement (which is the same order with respect to model performance because the more complex models are associated with better likelihoods), but now in a much less technical manner, and use AIC and BIC to inform a ranking/recommendation about models for prediction in modified habitat types. This part of the results section has been dramatically shortened (four paragraphs instead of eleven), with details of the models still available in the methods section and likelihood values in Supplementary Table 3. We have also added a new column to Table 1 (P28) that describes the application of each model to help the reader determine which models may be useful in practice.

2. I still find a coherence problem with the way abundance is estimated in non-neutral models (e.g. complete model). Obviously, there is a conceptual issue with first estimating abundance assuming interactions are driven by mass-action only, and then introducing species-specific preferences in the likelihood computation. This issue is however not only conceptual, as the approach could bias the estimates and alter the final model comparison. Imagine for instance that a given host species is highly vulnerable because of its traits (e.g. lack of defenses). The estimated relative abundance of this species will be high in the mass-action model and as a consequence, this will lower the estimated gamma values in the complete model, much below the 'reality' (it will basically tend to 1). It poses obviously a problem for the interpretation of the gammas, but also it could bias the inference in the altered habitat. Imagine for instance that abundance of that species is reduced considerably in the altered habitat. The predicted interaction probabilities, across all parasitoids, will consequently decrease much more than they should if the estimated gamma were much larger than 1. As a result, the fit of the 'complete model' in the altered habitat will appear much worse than it should if the relative abundance was appropriately estimated.

The bottom line is that the abundance should be estimated simultaneously as the preference to prevent biases.

The reviewer raises a very nuanced point about how the effective abundances and interaction preferences that we propose should be understood. Before we present a more detailed discussion on the topic, we would like to highlight the additions that we have made to the manuscript in direct response to the reviewer's insightful comment. First, we have added a fuller discussion of the assumptions made when estimating relative species abundances using our approach (P15L311). Second, we have added a discussion of potential ways of clarifying the role of species behaviour in determining network structure (P15L321).

If we may summarise the reviewer's example: Consider a host species with poor defences that consequently receives many counts (interactions) from multiple parasitoid species. Due to the many counts recorded to the host species, the value of the effective abundance estimated for this host will likely be relatively large. This relatively large effective abundance may then lead to lower interaction preferences than we think the host ought to have given its poor defences. When using the interaction preferences associated with this host at a new location with a different distribution of relative species abundances, we may obtain predicted interaction counts that are lower than they perhaps should be.

This example actually raises two separately interesting questions: (i) How does estimating relative species abundance affect the reliability of interaction preferences and therefore predictions? and (ii) How are interaction preferences related to species' biological characteristics?

Before we directly address the two questions, it is important to stress that we *do* always estimate abundance simultaneously with interaction preferences. (We had not previously made this important point in the main text, so we thank the reviewer for bringing this potential confusion to our attention; we have added clarifying remarks in the Methods section, P20L436, with details in Section 2.4 of *Supplementary Information*, pages 26 and 27.) What we also often do though, is combine the interaction preferences from a calibration (training) data set with the

estimated abundances in a test data set to make predictions about ecological network structure. This is due to lack of data on independent measurements of relative species abundance, and is a point we return to when discussing question (i). It is also worth noting that the example given by the reviewer may not necessarily lead to interaction preferences tending to 1 as a host species receives more and more interaction counts. This is because the final value for an interaction preference between a given host and parasitoid species depends on how many other interactions the two species are also involved in, and in what relative frequencies. Depending on the exact distribution of interaction counts among species in the community, the host species in the example may indeed be associated with interaction preferences greater than 1. Nevertheless, this aside does not get to the heart of the reviewer's point.

What is clear about our approach to estimating relative species abundances (effective abundances) and interaction preferences is that it will tend to "force" an explanation in terms of mass action. This is essentially the reasoning behind the reviewer's example: without additional information, our approach will attribute to mass action as much explanation of empirical data as possible, potentially at the expense of under-estimating "inherent" interaction preferences (below, we discuss what we mean by "inherent"). Clearly, this is not ideal. But given the current lack of data, we prefer to use this conservative method that tends to under-attribute changes in network structure to changes in species behaviour, rather than the opposite case which would lead to potentially more false-positive attributions to behaviour.

Having just described the tendency for our approach to favour an explanation in terms of mass action, we now discuss how this affects our ability to use interaction preferences to make predictions of network structure, i.e., question (i). In our work, we show that interaction preferences from one habitat type can be used to successfully predict network structure in *similar* habitat types (using the alternative preferences model). This is despite there being different distributions of relative species abundances between the habitat types. This strongly suggests that interaction preferences do reflect some biological/ecological features of a system that, moreover, can be useful for prediction. We also show that interaction preferences can change between *different* habitat types. What is currently difficult to determine, though, is how much any change in a given interaction preference is due to a more extreme change in relative species abundance (compared to changes between similar habitat types) and/or changes in environment (e.g., due to easier host hiding in more complex habitats with greater tree coverage). At present, both of these influences are rolled into our interaction preferences. We show that these interaction preferences can still be used to make decent predictions about network structure in modified environments (using the correlated preferences model etc.), but the next step is to try and tease apart precisely which species-level and environmental features cause observed changes in interaction preferences.

A more detailed investigation of interaction preferences is necessary to begin adequately answering question (ii). As suggested above, our interaction preferences currently confound two aspects: "inherent" preferences between parasitoid and host, and complicating factors due to environment. By "inherent" preferences, we mean some kind of baseline expectation for how often a parasitoid species would select a particular host, given a choice of multiple alternative hosts (for networks we think of this baseline selectivity at the population level rather than the individual level). We also discuss "inherent" preferences in the context of potential and realised niche in *Supplementary Information* on page 37. In our opinion, the best way of determining "inherent" preferences is using laboratory experiments. Although usable results are a few years away, we are actively developing experimental protocols to help inform baseline expectations

of host selectivity. Once an expectation is established, then one is better able to assess the effect of environment. For example, whether habitat complexity masks “inherent” preferences and therefore makes a system more likely to (appear to) follow mass-action-like processes. Our current work offers a number of exciting generalisable hypotheses that could be tested with new data and further study.

This is all to say that the reviewer’s comment and example prompted much reflection on our work. We now touch upon some of these points in the main text, which we hope will help the reader understand both the power and limits of our approach.

3. I appreciate the efforts to compare the different distribution and a summary of this discussion should find its place somewhere in the methods. But I must say I remain skeptical of the multinomial model. The argument that the multinomial is a generalization of the binomial distribution is irrelevant in the actual context, because ecologically speaking they describe two totally different stochastic processes. On the one hand, the binomial model represents the probability that an interaction occurs if you present a host to a given adult parasitoid, whatever is the community composition. This stochastic process describes for instance the probability that an egg laid in the host will develop into an adult, which has nothing to do with community composition. On the other hand, the multinomial model represents the probability that the parasitoid picks a given host, knowing all other potential hosts. The multinomial model has the advantage of accounting for higher order interactions (e.g. if there is active foraging), but this could also be accounted for in a conditional binomial model. Further, there should be a multinomial model for each community composition.

As suggested by the reviewer (final comment), we now describe the stochastic processes represented by the multinomial distribution, as well as a justification for its use, its hidden assumptions, and its limitations (P16L332). We also compare and contrast our use of the multinomial distribution for calculating model likelihoods to the binomial distribution. Furthermore, as suggested, we now include a detailed description and discussion of other possible distributions that could be used to calculate likelihood, along with further comparison of the multinomial and binomial distributions in *Supplementary Information* (pages 32 to 37; wherein we directly address the fact that there is a distinct multinomial distribution for each community composition, P34L476). Although this discussion does not impact our main methodological contribution of separating interaction preferences from relative species abundances, it is nevertheless an important aspect of modelling ecological networks that is now clearer, and, through our broader discussion, more useful and informative to the reader. We thank the reviewer for stimulating correspondence that has improved the rigour and relevance of our work.

It remains to be proven, I agree, but as I said my intuition is that the interaction between two species is a pairwise phenomenon, not a community one. Any individual parasitoid is not sampling the entire community before setting into an host. Likely, the adult will attempt oviposition on the first host it finds. The development of the larvae will depend on the quality of the host, irrespective of the composition of the community. Or may be, with enough information, we would find it is a mixture of the two.

This is a very nice ecological interpretation of the difference between using the multinomial or binomial distribution to calculate likelihood. We have incorporated this idea in our new discussion of the topic. We have also added an ecological example in which pairwise interactions may be a community phenomenon (P16L334): “[The multinomial distribution represents] the probability that a parasitoid picks a given host, conditioned on information about other hosts in the community. This conditioning is necessary if, for example, the abundances of particular host species lead to parasitoids forming search images [ref 28] that affect their per capita probabilities of attacking other hosts in the community.” Although we may differ with the reviewer on emphasis, we fundamentally agree with his general point and end our discussion of the topic with (P16L343): “The binomial distribution assumes that network structure is primarily a pairwise phenomenon, whereas the multinomial distribution assumes that it is primarily a community phenomenon, and likely it is a mixture of the two.”

A numerical example will better illustrate my point. Consider a community with host abundances $x_i = \{1, 6, 3\}$ and relative preferences $\gamma = \{1, 2, 0.5\}$. The interaction probability with the first host will be $1x_1 / (1x_1 + 6x_2 + 3x_3) = 1/14.5 = 0.069$. If for whatever reason, the third host species is absent at another location, this interaction probability will change to $1x_1 / (1x_1 + 6x_2) = 1/13 = 0.077$. This might be reasonable if the total number of larvae remains the same and we have to spread them among a reduced set of hosts. This is unlikely to be the reality, but it could be a reasonable assumption for host-parasite interactions. I am far from convinced however that the same reasoning would apply to other types of interactions where the number of plant visits or killing events by predators is not constant.

We appreciate the example and see the reviewer’s point. Indeed, it is one of the reasons why we focused on host-parasitoid systems and data sets that each has a similar set of species across modified and unmodified habitat types. However, we now acknowledge and discuss why using a multinomial distribution to calculate likelihoods may require more careful thought for other types of ecological network (*Supplementary Information*, P34L485).

Pushed to the extreme, the problem with the multinomial model is that the interaction probability between a given pair of host and parasitoid will converge to 1 as other species are removed from the community. The interaction then becomes deterministic.

We agree, and include this point when discussing interpretations of the multinomial and binomial distributions for calculating likelihoods (see *Supplementary Information*, P34L481).

All of this raise concerns in the interpretation of what is exactly the stochastic process the model is describing. My fear is that actually, the two processes above described (ie the pairwise interaction and the selection function) are mixed. Perhaps the authors are clear about this, but it is not in the main text and will obviously not be for the reader that has never thought about interactions as a probabilistic process.

We hope that the new discussion in the main text, along with a more detailed description of other probability distributions for calculating likelihoods in *Supplementary Information*, will not only help the reader better understand what we have done, but also serve as a more general introduction to thinking about interactions as a probabilistic process. In particular, at the reviewer's suggestion, we now emphasise the different assumptions about stochastic processes underlying the multinomial and binomial distributions, and how the two distributions may relate under our current approach (see *Supplementary Information*, P33L464).

To be more constructive, I would suggest the authors to better describe what is the stochastic process they represent, and consequently why they choose a multinomial model. Finally, the hidden assumptions of this decision should be clearly explained.

Again, we thank the reviewer for his constructive comments and now include the suggested discussion in the main text, as described above.

REVIEWERS' COMMENTS:

Reviewer #4 (Remarks to the Author):

I would like to apologize for delaying my evaluation. I knew it would be impossible to do it faster when I accepted the invitation, but I wanted to have a look at the latest revisions.

Again, I would like to thank the authors for the great attention they gave to my several comments. I think the manuscript greatly improved throughout revisions, both in terms of scientific rigor and of accessibility. I wish the authors the paper will attract attention and I would like to congratulate them for this great contribution.

I still have opinions about the science, but these will require future research and will not be solved through the review process. I signed my first review, so I will be happy if the authors contact me and want to keep the conversation going.

Response to reviewer comments

Reviewer comments are quoted in full in typewriter font, and each point is followed by our response.

Reviewer 4

I would like to apologize for delaying my evaluation. I knew it would be impossible to do it faster when I accepted the invitation, but I wanted to have a look at the latest revisions.

Again, I would like to thank the authors for the great attention they gave to my several comments. I think the manuscript greatly improved throughout revisions, both in terms of scientific rigor and of accessibility. I wish the authors the paper will attract attention and I would like to congratulate them for this great contribution.

I still have opinions about the science, but these will require future research and will not be solved through the review process. I signed my first review, so I will be happy if the authors contact me and want to keep the conversation going.

We would again like to thank the reviewer (Dominique Gravel) for his continued support and constructive criticism of our work. We agree that the manuscript has improved throughout revisions, in large part due his thoughtful comments and suggestions. We thank him in acknowledgements and look forward to continuing the conversation directly.